# Efficient molecular doping of polymeric semiconductors improved by coupled reaction

Jiahao Pan[1], Jing Wang[1,2], Kuncai Li[1], Xu Dai[1], Qing Li[3], Daotong Chong[2], Bin Chen [2], Junjie Yan[2] & Hong Wang [1,2] ✉

Exploring chemical doping method to improve the electrical conductivity of polymers is still very attractive for researchers. In this work, we report a developed method of doping a polymer semiconductor aided by the coupled reaction that commonly exists in biological systems where a non-spontaneous reaction is driven by a spontaneous reaction. During the doping process, the chemical reaction between the dopant and the polymer is promoted by introducing a thermodynamically favorable reaction via adding additives that are highly reactive to the reduction product of the dopant to form a coupled reaction, thus significantly improving the electrical conductivity of polymers by 3–7 orders. This coupled reaction doping process shows the potential of wide applications in exploring efficient doping systems to prepare functional conducting polymers, which could be a powerful tool for modern organic electronics.

Redox molecular doping has been widely used to control the electrical properties of polymeric semiconductors, which is a powerful tool to improve the performance of many organic devices such as field-effect transistors[1], light-emitting diodes[2], solar cells[3], as well as thermoelectric devices, etc[4]. Molecular doping occurs through transferring electrons from the polymeric semiconductors to the added molecular dopants (for example, p-type doping). Although the electron-transfer process is very complex that can be affected by many factors such as the dopant-polymer miscibility[5], the packing order of the polymer chains[6], etc. In many cases, the electron affinity (EA) and the ionic potential (IP) values are still used as an important indicative to determine whether the efficient electron transfer (doping) occurs between the dopant and the polymer[7]. Many researchers believe that dopant with a larger EA than the IP of polymer will be favorable for efficient electron transfer from the polymer to the dopant in p-type doping[1,8,9]. Watanabe et. al. claims that the EA of the dopant should match or exceed the IP of the polymer for p-doping[10]. Moulé et al. suggests that the EA of the dopant must be higher than the IP of polymer by ~0.11 eV for the efficient electron transfer to occur[8].

In the past years, tuning the electron affinity of dopants to promote efficient p-type doping in polymeric semiconductors has been successful in many cases[11,12]. However, increasing the electron affinity often causes chemical instability for both the dopants and the doped polymeric materials which has become a major challenge to extend the scope of potential molecular dopants[10]. Attempts have been placed on exploring strategies to mitigate this problem by tunings Coulomb interactions between the doped polymeric semiconductors and the counter ions[10,13,14] or lowering the activation energy of the doping reaction with catalysts[7], etc.

Using additives to activate the dopants for higher doping efficiency is a commonly used strategy. However, doping mechanisms are far from being well-understood. Exploring mechanisms for the ternary system doping to improve the doping efficiency is still very interesting to researchers. A few related works have been reported lately[7,14]. For example, Facchetti and Guo et. al. reported that the n-doping efficiency of organic semiconductors by the n-type dopant, N-DMBI-H, could be improved by the transition metal nanoparticles additives such as Pt, Au, and Pd, which acted as the catalysts that promoted the

[1]State Key Laboratory of Multiphase Flow in Power Engineering & Frontier Institute of Science and Technology, Xi'an Jiaotong University, Xi'an, China. [2]School of Energy and Power Engineering, Xi'an Jiaotong University, Xi'an, China. [3]College of Chemistry and Chemical Engineering, Dezhou University, Dezhou, Shandong, China. ✉e-mail: hong.wang@xjtu.edu.cn

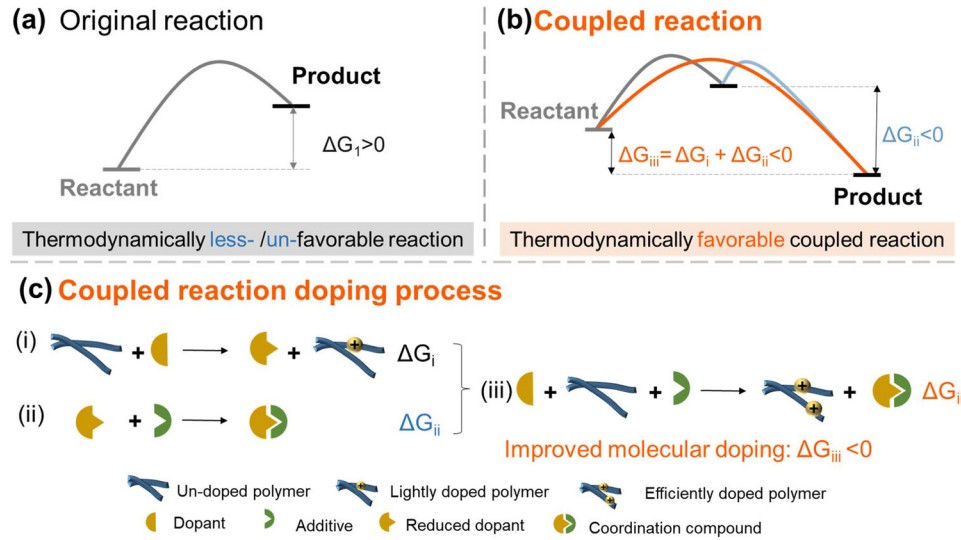

**Fig. 1 | Illustration of different doping process.** The original reaction (**a**), the coupled reaction (**b**), and the coupled reaction doping process (**c**).

doping reaction[7]. Yamashita et. al. reported that the p-doping efficiency of organic semiconductors by the p-type dopant, benzoquinone, could be promoted by ionic liquid additives such as bis(trifluoromethyl sulfonyl) imide (TFSI) and bis(nanofluorobutanesulfonyl) imide (NFSI), which provided counter anions to stable the oxidized organic semiconductor[15]. Watanabe and Sirringhaus et. al. reported that the p-doping efficiency of organic semiconductors by the p-type dopant, 2, 3, 5, 6-tetrafluoro-7, 7, 8, 8-tetracyanoquinodimethane (F4TCNQ), could be improved by the ionic liquid additives such as Li-TFSI, BMP-TFSI, and TBA-TFSI, etc. through a ion-exchange mechanism[10,14].

In addition to these recently reported methods, using coupled reactions is a theoretically promising way to improve the doping level of polymeric semiconductors with no need to increase the electron affinity of molecular dopants. It commonly exists in biological systems where a non-spontaneous reaction is driven by a spontaneous reaction[16], which, in chemistry, often represents the one that a thermodynamically unfavorable (Gibbs energy change, $\Delta G > 0$) reaction is driven by the thermodynamically favorable ($\Delta G < 0$) reaction by coupling or mechanistically joining the two reactions via a shared intermediate. In principle, the efficient molecular doping of polymeric semiconductors can be promoted by introducing a thermodynamically favorable reaction via adding additives to turn the doping process into a designed coupled reaction. However, coupled reaction doping still lacks study for organic semiconductor doping in previous works.

Here we demonstrate that using coupled reaction can successfully overcome electron-transfer limitations in the Marcus theory to have efficient molecular doping of polymeric semiconductors. We focus on the chemically p-type doping of the recently reported bipolar donor-acceptor copolymer poly (2, 5-bis(2-octyldodecyl)-3, 6-di(thiophen-2-yl)diketopyrrolo[3, 4-c]pyrrole-1, 4-dione-alt-thieno[3, 2-b]thiophen) (DPP4T) with the p-type dopants of nitroxide derivatives. To improve the doping level of DPP4T, Lewis acid additives have been added to form a coupled reaction since the added additives thermodynamically prefer to combine with the reduced nitroxide intermediate. It turns the thermodynamically less-/un-favorable doping reaction (Fig. 1a) into a thermodynamically favorable one (Fig. 1b), which subsequently results in efficient molecular doping of the polymers as illustrated in Fig. 1c.

Moreover, we extended the coupled reaction doping process to the nitroxide derivatives containing nitroxide cations that are capable of accepting two electrons per dopant. It avoids the complex synthesis of the dimeric version of common dopants in the recently explored

avenue to have two-electron transfer for one dopant[17,18]. The Lewis acid-assisted coupled-reaction doping process involves a two-electron transfer step from the polymer to the dopant, similar to that of the double doping method reported by Muller et. al.[19], which may has less effects on the packing structure of polymer films than the single-electron transfer doping (one electron per dopant) in literature due to the less volume/weight fraction of dopants addition[17,18,20–23].

The coupled-reaction doping results in significant improvements in the electrical conductivity of DPP4T films by several orders in magnitude. Besides, unusual n-type properties are observed while the films are heavily p-doped by using the coupled reaction doping process, similar to the FeCl₃ doped DPP4T reported by Wang[24,25] and Graham[26]. A series of p-type dopants are developed for the preparation of bi-polar conducting polymers with the coupled reaction strategy. The results demonstrate that coupled reaction doping is a powerful tool in both improving the electrical properties and tuning the carrier polarity of organic semiconductors for modern organic electronics.

## Results and discussion

We used a commonly reported drop-casting method to prepare the doped organic films (Fig. 2a). The well-mixed solution was drop-casted on glass substrates and then dried under $N_2$ protection at room temperature. Detailed preparation process was described in the Supplementary Information. The molecular structures of the polymeric semiconductors, the dopants, and the Lewis acid additives are shown in Fig. 2b.

The electrical properties of the doped DPP4T film were measured. DPP4T film has a higher electrical conductivity when it is doped with the coupled reaction method (doped by the dopant and the additional additives together). As shown in Fig. 3a, the electrical conductivity of the DPP4T film doped by 2, 2, 6, 6-tetramethylpiperidinooxy (TEMPO, D1) is in the range of $<10^{-6}$ S/cm. While adding D1 together with the additive tris(pentafluorophenyl) borane (BCF, A1) at the ratio of D1:A1 = 1:1, the electrical conductivity of the D1A1 doped DPP4T (DPP4T-D1A1) film increases quickly at the dopant molar ratio of 0.9 (D1A1: DPP4T aromatic ring = 0.9). The electrical conductivities were 0.01 and 0.07 S/cm for DPP4T-D1A1 at the dopant molar ratio of 0.01 and 0.05, respectively. However, the Seebeck coefficient was not measurable when the dopant molar ratio of D1A1 was <0.1. The electrical conductivity and the Seebeck coefficient of DPP4T-D1A1 at the dopant molar ratio of 0.2 were 3.4 S/cm and 156 µV/K, respectively (Fig. 3b). The maximum electrical conductivity reaches 15.5 S/cm at the dopant molar ratio of 0.9, which is several orders in magnitude higher

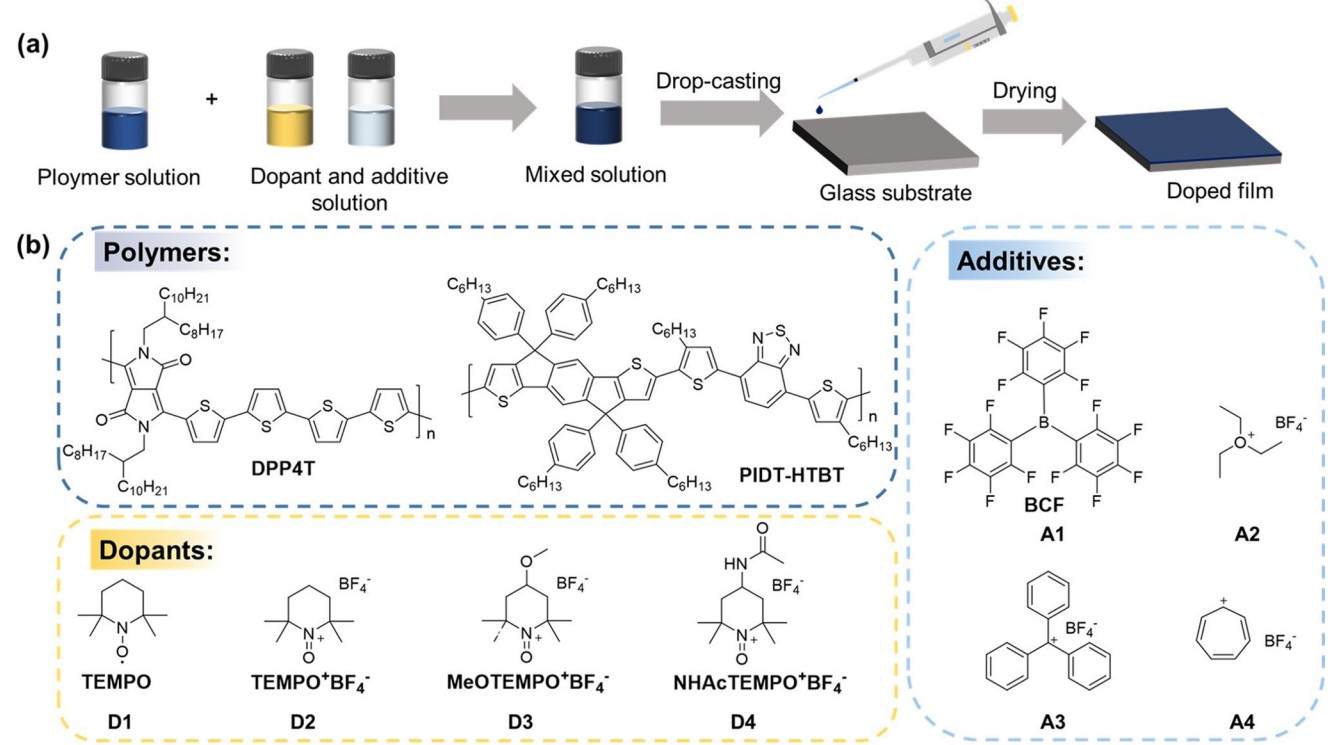

**Fig. 2 | Experimental illustration of the coupled reaction.** Doped film fabrication process (**a**), chemical structures of polymers, dopants, and additive used in this work (**b**).

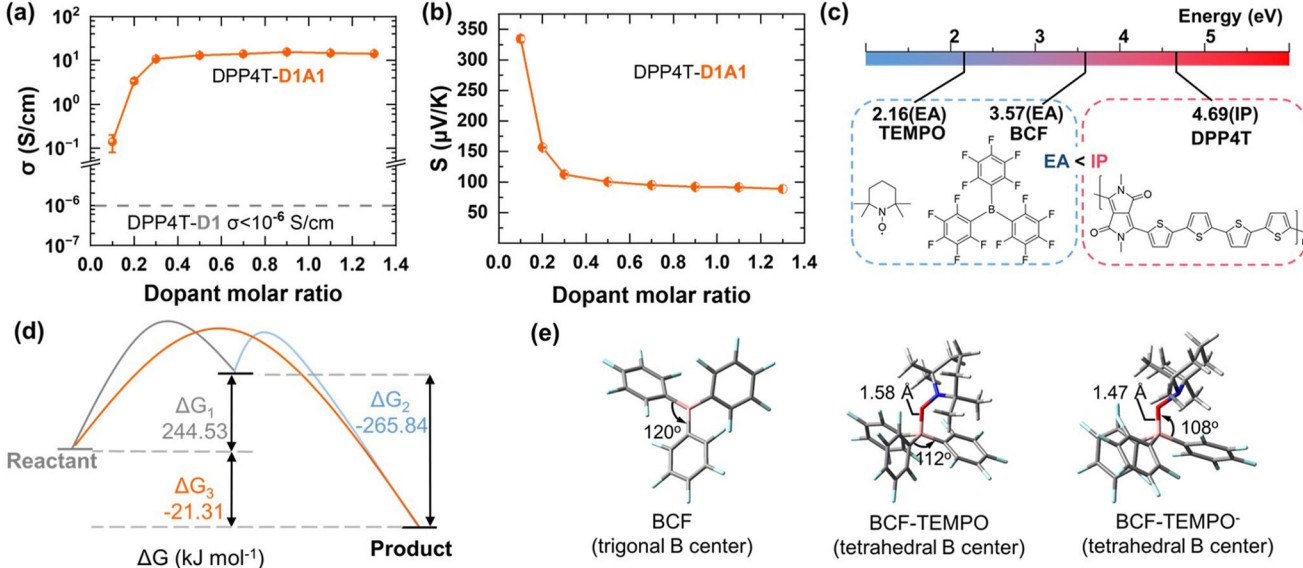

**Fig. 3 | Coupled reaction doping of DPP4T-D1A1.** Electric conductivity (**a**) and Seebeck coefficient (**b**) of DPP4T-D1A1 as a function of the dopant molar ratio. Comparison of EA values of dopant D1 and additive A1 with IP value of DPP4T (**c**). Calculated Gibbs free energy change (ΔG) in coupled reaction doping process of DPP4T-D1A1 (**d**). Theoretically optimized structures of BCF, BCF-TEMPO, and BCF-TEMPO· with DFT calculation (**e**). Error bars were standard deviations from at least 3 sample.

than that of the DPP4T-D1 film. After reaching the maximum values, the electrical conductivity of DPP4T-D1A1 decreases a little bit.

The DPP4T-D1A1 films exhibit a high p-type Seebeck coefficient of +334.6 μV/K which decreases with the increase of dopant concentration (Fig. 3b). It reaches a platform at +94.9 μV/K while being heavily doped by D1A1. The positive Seebeck coefficient demonstrates that the DPP4T-D1A1 film is a p-type material. Nevertheless, the Seebeck

coefficient of DPP4T-D1 film is not measurable due to its extremely poor electrical conductivity.

To understand the role of the Lewis acid additive A1 in improving the electrical conductivity of the doped DPP4T films, the electron affinity EA of the dopants and the ionic potential IP of the polymer were calculated by using Gaussian 09 at the level of the B3LYP hybrid functional. Detailed information was described in the Supplementary

Information and Supplementary Data 1. It should be noted that both the EA and the IP values were only used for qualitatively understanding the doping process since these values were calculated with a single molecule or oligomer under ideal environmental conditions without considering other effects such as molecular packing and molecular proximity, etc[27]. Therefore, the theoretical EA and IP values were proposed as indicative only due to the limitation in accuracy from practical conditions. Figure 3c shows the EA values of dopants and the IP value of DPP4T. The dopant D1 has a small EA of 2.16 eV, which is smaller than the value of the IP 4.69 eV of DPP4T. In the original reaction doping process (DPP4T doped with dopants only), it is unfavorable for electrons transferring from the polymer to the dopant according to the Marcus theory, which then leads to the extremely poor electrical conductivity of DPP4T-D1 in Fig. 3a. The EA of A1 has also been calculated to exclude the direct oxidation of DPP4T by A1. The calculated results demonstrate that the reaction between DPP4T and A1 is also unfavorable since the EA of A1 is lower than the IP of DPP4T as well. The calculated method used here is acceptable since the obtained EA of A1 is 3.57 eV which is close to the value reported in literature 3.03–3.31 eV[28]. The IP of DPP4T and the EA of A1 and D1 were also obtained experimentally, which matched the calculated values well (Supplementary Figure S1 and Supplementary Table S1). Detailed discussion was provided in the Supplementary Information. The unfavorable doping reaction results in poor electrical conductivity in the range of 0.05–0.12 S/cm for the DPP4T film doped by the additive A1 only (DTT4T-A1, Supplementary Figure S2). The DPP4T-A1 showed a low electrical conductivity, which should be due to the slightly doping of DPP4T by A1 since A1 was a well-known Lewis acid that might form a water-Lewis acid or other complex to promote the p-type doping of DPP4T as suggested by Nguyen and Jang et. al.[29,30]. We used F4TCNQ(2,3,5,6-Tetrafluoro-7,7,8,8-tetracyanoquino dimethane) instead of A1 as an additive in the doping process. It was found that DPP4T-D1F4TCNQ had a poor electrical conductivity (Supplementary Figure S3). The results indicated that A1 played an important role in the coupled reaction doping process. In the meanwhile, it indicated that the electrical conductivity would not be increased by simply increase the molar ratio of dopants with low EA. The results also demonstrate that both the dopant only and the additive only cannot lead to efficient molecular doping of DPP4T films for high electrical conductivity.

We further calculated the Gibbs free energy change (ΔG) of the related reactions to understand the significant improvement of the electrical conductivity in DPP4T-D1A1 films (Fig. 3c). Detailed information is described in the Supplementary Information. For the original doping reaction process, the Gibbs free energy change is calculated to be $\Delta G_1 = 244.53$ kJ·mol$^{-1}$ as shown in Eq. (1):

$$DPP4T + D1 \rightarrow DPP4T^+ + D1^- \quad \Delta G_1 = 244.53 \, kJ \cdot mol^{-1} \quad (1)$$

where the DPP4T$^+$ is the cation when DPP4T loses an electron and the D1$^-$ is the anion when D1 receives an electron. The positive $\Delta G_1$ shows that the original doping reaction is thermodynamically unfavorable. After A1 is added, it forms a complex with D1$^-$ as shown in the Eq. (2):

$$A1 + D1^- \rightarrow A1 \cdot D1^- \quad \Delta G_2 = -265.84 \, kJ \cdot mol^{-1} \quad (2)$$

The formation of the complex A1·D1$^-$ is more favorable than the formation of A1·D1. Figure 3e shows the theoretically optimized structure of the complexes A1·D1 and A1·D1$^-$, which demonstrate that A1·D1$^-$ is more stable than A1·D1. The original A1 is a planar triangle configuration with boron B at the center according to the Valence Shell Electron Pair Repulsion Theory (VSEPR)[31]. Density functional theory (DFT) calculation indicates that the angle for the carbon-boron-carbon bonds (C-B-C) in the original A1 tris(pentafluorophenyl) borane is 120°. After coordinating with D1 and D1$^-$, the angles of the C-B-C become 112° and 108° in the complexes A1·D1 and A1·D1$^-$, respectively. The

formation of the coordination B-O bond turns the tri-coordinate B into the quadri-coordinate B whose ideal bond angle is 109.5°. The C-B-C bond angle in A1·D1$^-$ is closer to the ideal bond angle, which indicates that A1·D1$^-$ is more stable than A1·D1. In addition, the coordination binding energy of A1·D1$^-$ is −587.63 kJ·mol$^{-1}$, which is about an order in magnitude higher than that of A1·D1(−62.39 kJ·mol$^{-1}$). In the meanwhile, the calculated coordination B-O bond in the optimized A1·D1$^-$ structure is 1.47 Å, which is close to the covalent B-O bond length in the range of 1.34−1.47 Å[32]. While the calculated coordination B-O bond for A1·D1 is 1.58 Å. The shorter the B-O bond is, the higher the bond energy it will have. The results demonstrate the ease of the formation of A1·D1$^-$ in the doping process. Experimental identification of the formation of A1·D1$^-$ has also been reported with X-ray crystallography data in previous works[33].

The idea of the formation of the coordination compound between the dopant and the additive was also supported by the following reasons: Firstly, the dopant D2 and the additive A1 were surrounded by each other since they were mixed thoroughly with the polymer in solution during the film preparation process as described in the original Supplementary Information. Secondly, the reduction products of D2 ions thermodynamically preferred to form a coordination compound with the Lewis acid A1 according to the theoretical calculation and the experimental results reported in previous works[34–36]. The formation of the single crystal of D2$^-$A1 in a complex solution system reported in previous works[33] strongly demonstrated the good coordination capacity between D2$^-$ and A1. At last, the electrospray ionization mass spectrometry (ESI-MS) results were shown in our original manuscript, which showed the existence of the molecular ion peak at m/z = 668 for D2$^-$A1 in Supplementary Figure S17 of the original Supplementary Information. Based on the above facts, it is reasonable to believe that the coordination compound of the dopant and the additive forms in during the film preparation process which subsequently induces the efficient molecular doping of the polymer.

Therefore, the whole reaction can be written as shown in Eq. (3):

$$DPP4T + D1 + A1 \rightarrow DPP4T^+ + A1 \cdot D1^- \quad \Delta G3 = -21.31 kJ \cdot mol^{-1} \quad (3)$$

Then, the addition of A1 turns the thermodynamically unfavorable doping reaction (Gibbs energy change, $\Delta G_1 > 0$) reaction into a thermodynamically favorable doping reaction ($\Delta G_3 = -21.31$ kJ·mol$^{-1} < 0$), which is called a coupled reaction as reported in the literature[16]. The coupled reaction doping process subsequently leads to a significant improvement in the electrical conductivity from <10$^{-6}$ to 15.5 S/cm for DPP4T-D1A1 (Fig. 3a).

To further improve the electrical properties as well as to check the wide applicability of this proposed coupled reaction doping method, the coupled reaction doping was then extended to the p-doping of DPP4T film with the salt of the nitroxide derivatives containing TEMPO cations, e.g. TEMPO$^+$BF$_4^-$ (D2) in Fig. 2b. One TEMPO$^+$ is capable of accepting two electrons as shown in Eq. (4)[37,38], which may minimize the disturbance of the packing structure of polymer films for higher electrical conductivity as suggested in literature[17–23]. Supplementary Figure S4 showed that DPP4T films doped by A1 and Bu$_4$N$^+$BF4$^-$ had a poor electrical conductivity as compared the DPP4T-D2A1. The results showed that TEMPO$^+$ played an important role in the coupled reaction doping process. The dopant TEMPO$^+$PF$_6^-$ (D5) was synthesized according to the literature[39,40] to check whether boron tetrafluoride anion is indispensable in the coupled reaction induced p-type doping process. Figure S5a showed that significant improvement could also be obtained while doping DPP4T with D5A1 (DPP4T-D5A1) with a maximum electrical conductivity of 34.5 S/cm and the polarity of Seebeck coefficient also changed from p- to n-type after increasing the dopant molar ratio up to >0.5 (Supplementary Figure S5b). The results were similar to that of DPP4T-D2A1, which demonstrated the wide compatibility of the coupled reaction doping process. We also found that

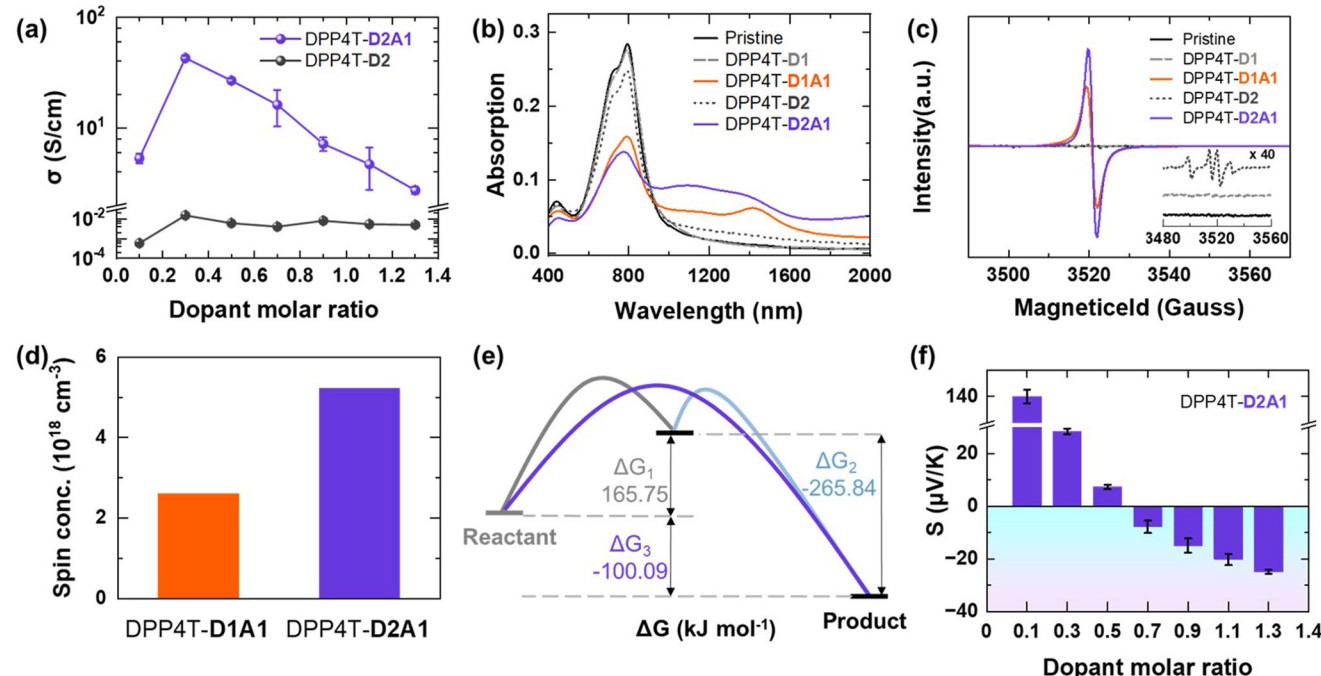

**Fig. 4 | Coupled reaction doping of DPP4T-D2A1.** Electric conductivities of DPP4T-D2A1 and DPP4T-D2 as a function of dopant molar ratio (**a**). UV–vis–NIR spectra (**b**) and ESR spectra (**c**) of undoped and doped DPP4T films. The spin concentration of DPP4T-D1A1 and D2A1 was obtained from quantitative ESR experiments (**d**). Calculated Gibbs free energy change (ΔG) in coupled reaction doping process of DPP4T-D2A1 (**e**). Polarity switching of DPP4T-D2A1 in Seebeck coefficient measurement as a function of dopant molar ratio (**f**). Error bars were standard deviations from at least 3 sample.

DPP4T-D2A1 reached the best doping result in dopant: additive molar ratio at 1: 1 (Supplementary Figure S6). The detailed screening of dopant-to-additive ratio was discussed in the Supplementary Information.

The Eq. (4) shows that TEMPO$^+$ accepts two electrons one by one to produce TEMPO$^-$ [37,38].

$$\text{TEMPO}^+ \underset{-e}{\overset{+e}{\rightleftharpoons}} \text{TEMPO}^\cdot \underset{-e}{\overset{+e}{\rightleftharpoons}} \text{TEMPO}^- \qquad (4)$$

where TEMPO$^\cdot$ is the radical status of TEMPO and TEMPO$^-$ is the TEMPO anion. The DPP4T-D2A1 films at the dopant molar ratio of 0.3 exhibited a good stability, which could maintain >85% of their electrical conductivity while being laminated with polyethylene terephthalate films for over 7 days (Figure S7). The temperature dependent electrical conductivity of DPP4T-D2A1 films was tested at the temperature range of 25 °C–120 °C. It indicated that the electrical conductivity maintained when the temperature was lower than 75 °C (Supplementary Figure S8a). The film preparation temperature was also optimized. It showed that DPP4T-D2A1 films prepared at room temperature exhibited the highest electrical conductivity which should be due to the poor stability of TEMPO$^-$ [41–43] at higher temperature (Supplementary Figure S8b).

We measured the electrical properties of DPP4T films doped by TEMPO$^+$ with/without the additive A1. Figure 4a shows the electrical conductivities of D2 and D2A1 doped DPP4T films as a function of the concentration of the dopants. The DPP4T-D2 film has a low electrical conductivity in the range of $10^{-3}$-$10^{-2}$ S/cm. Although the electrical conductivity of DPP4T-D2 film is low, it is much higher than that of DPP4T-D1 film since the D2 is a better p-type dopant with higher EA of 5.53 eV than D1 (EA = 2.16 eV) in theory. After adding A1, the DPP4T-D2A1 film exhibits a high electrical conductivity of up to 42.7 S/cm, which is about 2–4 orders in magnitude higher than that of the DPP4T-D2 film. It is also higher than that of the DPP4T-D1A1 films (0.1 - 15.5 S/cm in Fig. 3a). Although the electric conductivity of ~40 S/cm for the

drop-casted thick films in micrometer scale was lower than that of ~500 S/cm for the state-of-the-art p-type polymer PEDOT:PSS, higher electrical conductivity may be achieved while reducing the thickness of the films as suggested in previous work[18,29,30]. It was noted that the electrical conductivity of PEDOT:PSS films was also not high (-1 S/cm) in the early reported works[44].

The higher electrical conductivity of DPP4T-D2A1 films is mainly attributed to the higher doping level derived from the two-electron transfer doping process. Figure 4b shows ultraviolet-visible (UV-vis) spectra of the pristine and the p-type doped conducting films at each maximum electrical conductivity. Pristine DPP4T film has a main absorption peak that appeared at ~800 nm which is assigned to the absorption of the π-π* transition of polymer conjugate structure[45]. The addition of D1 has little effect on the UV-vis spectrum of DPP4T (Supplementary Figure S9), which is mainly due to the lack of p-type doping to DPP4T since the EA of D1 is much lower than the IP of DPP4T, leading to the extremely low electrical conductivity of <$10^{-6}$ S/cm (Fig. 3a). After the addition of A1, the absorption peak for π-π* transition decreases and a peak appears in the range of 1000–1600 nm for the DPP4T-D1A1 film. The peak is assigned to the formation of polaron which indicates the p-type doping of DPP4T film is promoted by the coupled reaction, thus leading to a high electrical conductivity of 15.5 S/cm. For the D2 and D2A1 doped DPP4T films, DPP4T-D2 film has a higher absorption peak of the polarons in the range of 1000–1600 nm than DPP4T-D1, indicating a better p-type doping[46] that is due to the higher EA of D2. After the addition of A1, the absorption peak of the polarons in the range of 1000–1600 nm increases compared with that of DPP4T-D1A1 film. In addition, a peak appears in the range of >1800 nm, which is due to the formation of bi-polarons. The higher absorption peak of polarons in DPP4T-D2A1 film demonstrates the more efficient p-type doping, thus resulting in higher electrical conductivity of DPP4T-D2A1 film (42.7 S/cm) compared to DPP4T-D1A1 film (15.5 S/cm). Although the additive A might induce structure disorder, the improved doping efficiency still resulted in the improvement of the electrical conductivity of the thick films

due to the increased carrier concentration (Hall measurement Supplementary Figure S16). The doping efficiency induced electrical conductivity improvement has also been reported in previous works[7,14,15] No obvious differences for polymer packing in DPP4T-D2A1 and DPP4T-D1A1 films have been observed in the Grazing Incidence Wide Angle X-Ray Scattering as shown in Figure S10.

The more efficient p-type doping in DPP4T-D2A1 film than that in DPP4T-D1A1 was further demonstrated by the electron spin resonance (ESR) measurement. Figure 4c and Supplementary Figures S11-S14 show the ESR spectra of the pristine and the doped conducting films at each maximum electrical conductivity. After being doped, the conducting polymer DPP4T generates unpaired electrons (polarons). The unpaired spin density in ESR spectra can be used to evaluate the doping level of conducting polymers[47]. Both the pristine DPP4T and DPP4T-D1 films in Fig. 4c (inserted image) have no peaks in the ESR spectra. Supplementary Figure S14 showed that both DPP4T-D1 and D1 only in solutions had strong signals which were assigned to the TEMPO radical[48–50], while DPP4T-D1 and D1 only dry films showed no signal. The results demonstrated that DPP4T would not quench TEMPO radical. It was hypothesized that the TEMPO had been evaporated in the dry films since the boiling point of TEMPO (193 °C) was close to that of the used solvent (179 °C). After the addition of A1, the DPP4T-D1A1 spectrum exhibits a clear peak at 3520 G, which demonstrates the generation of polarons by p-type doping of D1A1. For dopants, D2 and D2A1 doped DPP4T film, the DPP4T-D2 film has a small ESR peak at 3520 G since D2 shows higher EA than D1 (Fig. 3b). After the addition of A1, DPP4T-D2A1 has a higher ESR peak than DPP4T-D1A1 as well, which indicates the more efficient p-type molecular doping in DPP4T-D2A1 films[51,52]. The spin concentration has been calculated from the double integral of the signal as shown in Fig. 4d and Supplementary Figure S13, which can only be used to qualitatively understand the doping level of polymers[53,54]. Because It was known that the ESR/EPR often underestimated the carrier density at high doping levels because Pauli-type magnetic susceptibility was often observed and the spin density corresponded only to the subset of spins in close vicinity to the Fermi level[10,47,52,53]. The obtained spin concentration from the ESR spectra is in the order of DPP4T-D2A1 > DPP4T-D1A1 > DPP4T-D2 > DPP4T-D1 (Fig. 4d). The trend is consistent with the trend for the doping level obtained in the UV-vis measurements (Fig. 4b). We further checked the doping level of DPP4T-D1A1 and DPP4T-D2A1 films by using Hall measurements (Supplementary Figure S15), which show the same trend with the ESR and UV-vis results. According to the carrier concentration in Supplementary Table S2, we can get that one repeated monomer unit has 0.01 and 1 charge in DPP4T-D1A1 and DPP4T-D2A1 films, respectively (Figure S16). Figure S15a showed that the carrier mobility of DPP4T-D2A1 film was in the range of 0.06–2.5 cm$^2$/(V·s), which decreased with the increase of the dopant molar ratio since the dopants were non-conductive materials. Then, it increased at heavily doping level with the dopant molar ratio of 0.9. The rise of the carrier mobility should be due to formation of the crystalline regions as suggested by Graham et. al.[26] Supplementary Figure S15b showed that the electrical conductivity of DPP4T-D2A1 increased with the decrease of the carrier mobility, which was also commonly reported in previous works since the electrical conductivity of polymeric materials was affected by both the carrier mobility and the carrier concentration that was related to the efficient molecular doping level[54–58]. Although these measurements may be overestimated or underestimated the real doping level of doped polymer[56], the trend is clear that: (1) the doping level in DPP4T-D2A1 is higher than that in DPP4T-D1A1 and (2) the addition of A1 is helpful to improve the doping level of DPP4T films.

The doping of DPP4T with D2A1 is demonstrated to be a two-electron doping process. ESR spectrum of DPP4T-D2 shows that there are 4 peaks (Fig. 4c inserted image and Supplementary Figure S12). The main peak at 3520 G is the peak for the polarons due to the p-type doping of DPP4T by D2. The position of the other three peaks matches

well with the ESR peaks for TEMPO$^-$, which proves the generation of TEMPO$^-$ in DPP4T-D2 film during the doping process. Therefore, the TEMPO$^+$ accepts one electron to turn to TEMPO$^-$ firstly and then accepts another electron to become TEMPO$^-$ as shown in Eq. (4) in the doping process of DPP4T with D2A1. The final reduction product in DPP4T-D2A1 is the same as that in DPP4T-D1A1 (D1A1$^-$, TEMPO$^-$-BCF), which is demonstrated by the electrospray ionization mass spectrometry (ESI-MS) negative scan results with the molecular ion peak at m/z = 668 in Figure S17 Besides, the p-type doping requires less than half of the dopants to reach the maximum electrical conductivity for DPP4T-D2A1 (dopant molar ratio of 0.3, Fig. 4a), which also supports the conclusion that the two-electron transfer doping process in DPP4T-D2A1 film since it requires much less dopants than the one-electron transfer doping process in DPP4T-D1A1 film (dopant molar ratio of 0.9, Fig. 3a).

The doping of DPP4T with D2A1 is also a coupled reaction-promoted doping process. The doping reactions and the related Gibbs free energy changes are shown as follows:

$$DPP4T + D2 \rightleftharpoons DPP4T^{2+} + D1^- \quad \Delta G_5 = 165.75 \, kJ\cdot mol^{-1} \quad (5)$$

$$DPP4T + D2 + A1 \rightleftharpoons DPP4T^{2+} + A1\cdot D1^- \quad \Delta G_6 = -100.09 \, kJ\cdot mol^{-1} \quad (6)$$

The reaction in Eq. (5) is a thermodynamically unfavorable reaction with a positive $\Delta G_5 = 165.75$ kJ·mol$^{-1}$. The addition of A1 can react with the reduced nitroxide intermediate D1$^-$ as shown in Eq. (2). Therefore, the total reaction in Eq. (6) has a negative $\Delta G_6 = -100.09$ kJ·mol$^{-1}$, which demonstrates that the coupled reaction turns the reaction in Eq. (5) into a thermodynamically favorable reaction, thus leading to the high electrical conductivity of DPP4T-D2A1. The Gibbs free energy diagram is illustrated in Fig. 4e.

In addition to the high electrical conductivity, it is interesting to note that the DPP4T-D2A1 films exhibit n-type Seebeck coefficient after being heavily p-type doped by D2A1. Figure 4f shows that DPP4T-D2A1 is p-type with a high positive Seebeck coefficient of +140.0 μV/K at a low dopant molar ratio of 0.1. The positive Seebeck coefficient then decreases with the increase of the dopant molar ratio, which becomes negative at the dopant molar ratio of 0.7, indicating that the polarity switching of DPP4T-D2A1 from p- to n-type when DPP4T is heavily doped by the p-type dopant D2A1. The maximum n-type Seebeck coefficient of DPP4T-D2A1 is −24.9 μV/K. The DPP4T-D2A1 films were demonstrated to be electronic conduction dominated materials as shown in Supplementary Figure S18 rather than ionic conduction dominated ones. The Nyquist plots of DPP4T-D2A1 films at the dopant molar ratio of 0.3 (p-type) and 1.3 (n-type) showed data regrouped around a point (Supplementary Figure S18a, like for an ideal resistance), indicating a dominant electronic conduction of these films[59,60]. Supplementary Figure S18b also showed that the normalized electrical conductivities for the films were constant up to 10$^4$ and 10$^3$ Hz for the dopant molar ratio of 0.3 (p-type) and 1.3 (n-type), respectively. Indicating that electronic transport was dominated over ionic effects since the increase of the electrical conductivity often occurred at a very low frequency (about 0.3 Hz)[61–64]. Therefore, the ionic effects on the Seebeck coefficient and the electrical conductivity in the films may be ignored. The polarity change should be attributed to the high doping level (shown in Fig. 4d and Supplementary Figure S16), which might move the Fermi level below the valence band of the polymeric semiconductor as suggested in previous literature[24,25,65]. Another mechanism for the polarity change should be to the electrons delocalized from the crystalline regions as suggested by Graham et. al.[26]. However, the mechanism of the polarity change of the Seebeck coefficient is still not very convincing because the doping mechanism is far away from being well understood which may be affected by a lot of factors such as the packing order of polymer molecules, polymer structure, morphology

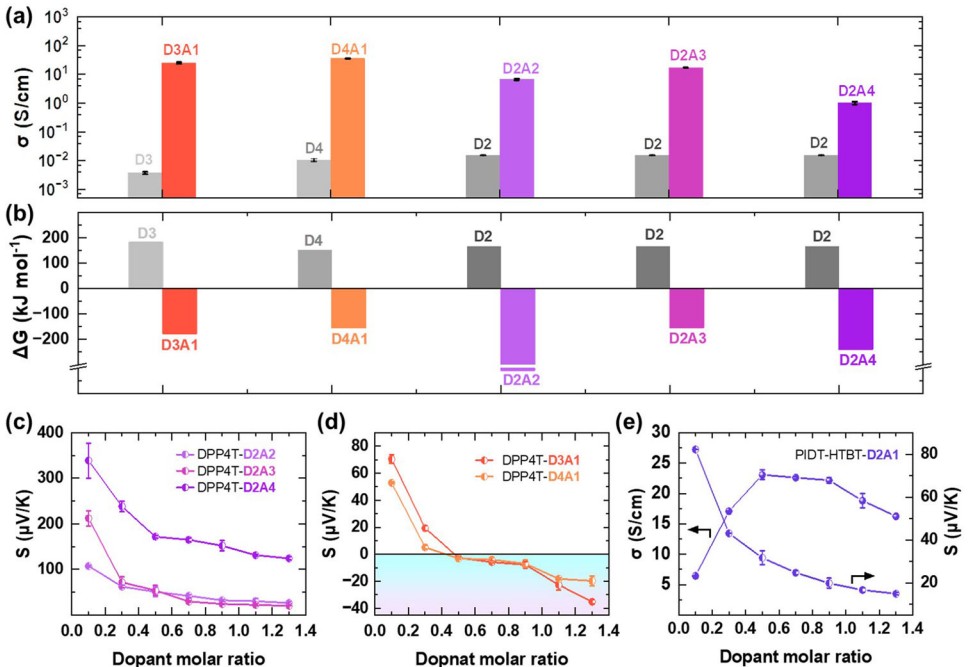

**Fig. 5 | Coupled reaction doping of other dopants, additives and polymer.** Electric conductivities (**a**) and Gibbs free energy change (**b**) of DPP4T films with different dopants; The Seebeck coefficients of DPP4T-D2A2, DPP4T-D2A3 and DPP4T-D2A4 as a function of dopant molar ratio (**c**); The Seebeck coefficients of DPP4T-D3A1 and DPP4T-D4A1 as a function of dopant molar ratio (**d**); The electrical properties of PIDT-HTBT-D2A1 as a function of dopant molar ratio (**e**). Error bars were standard deviations from at least 3 sample.

and dopant distribution, the size and structure of the dopant, etc. Further work is still undergoing to fully understand this phenomenon.

We further checked the wide applicability of the coupled reaction doping of DPP4T by D2 with different additives (A2, A3, A4) dopants as well as different nitroxide derivative dopants (D3, D4) with A1. Figure 5a shows that the electrical conductivity increases significantly after using the coupled reaction doping method. Detailed electrical conductivity as a function of the dopant molar ratio has been shown in Supplementary Figures S19, S20. The maximum electrical conductivity of DPP4T-D2 film is only ~$10^{-2}$ S/cm, which increase to 6.74 S/cm, 17.28 S/cm, and 1.02 S/cm for DPP4T-D2A2, DPP4T-D2A3, DPP4T-D2A4 films, respectively. The coupled reaction doping is also demonstrated to be suitable for other nitroxide derivative dopants (D3, D4) to improve the doping efficiency. The electrical conductivities for DPP4T-D3 and DPP4T-D4 films are ~$3.5 \times 10^{-3}$ S/cm and ~$10^{-2}$ S/cm, respectively. While the maximum electrical conductivities for DPP4T-D3A1 and DPP4T-D4A1 films reach up to 25.18 S/cm and 36.03 S/cm, respectively.

The Gibbs free energy change was calculated to confirm the coupled reaction doping process (Fig. 5b). For the doping reaction in DPP4T-D2, DPP4T-D3, and DPP4T-D4, the $\Delta G$ values are 165.75 kJ·mol$^{-1}$, 182.19 kJ·mol$^{-1}$ and 151.83 kJ·mol$^{-1}$, which are all positive, indicating the doping reactions are all thermodynamically unfavorable. While the $\Delta G$ values for DPP4T-D2A2, DPP4T-D2A3, DPP4T-D2A4, DPP4T-D3A1, and DPP4T-D4A1 are −509.36 kJ mol$^{-1}$, −153.00 kJ·mol$^{-1}$, −238.80 kJ·mol$^{-1}$, −176.85 kJ·mol$^{-1}$ and −154.69 kJ·mol$^{-1}$, respectively. The negative values indicate that the addition of the additives turns the thermodynamically unfavorable doping reactions into thermodynamically favorable doping reactions.

We measured the Seebeck coefficients of the coupled reaction doped DPP4T films by D2 and different additives (A2, A3, A4) dopants as well as different nitroxide derivative dopants (D3, D4) with A1. FigS. 5c, d show the Seebeck coefficients as a function of the dopant molar ratios. Different from the DPP4T-D2A1 films, the Seebeck coefficients of DPP4T-D2A2, DPP4T-D2A3, and DPP4T-D2A4 films are all positive when the dopant molar ratio is even up to 1.3. The Seebeck

coefficients of DPP4T-D3A1 and DPP4T-D4A1 films show a similar trend when increasing the dopant molar ratio, which can turn from positive to negative. The results demonstrate that the polarity changing of DPP4T-D3A1 and DPP4T-D4A1 films occurs when they are heavily p-type doped, which is similar to DPP4T-D2A1. The polarity changing may be also related to the EA or the size of the dopants as well as the packing of polymer chains[24–26]. Further work is still required to understand this interesting polarity change in heavily p-type doped conducting polymer films.

In addition to doping DPP4T, the coupled reaction doping strategy can also be applied to another polymer semiconductor, such as indacenodithiophene (IDT), a commonly used polymer for solar cells[66]. Fig. 5e shows the electrical properties of poly (indaceno[1,2-b:5,6-b']dithiophene-4,7-bis(4-hexylthiophen-2-yl)benzo[c][1,2,5] thiadiazole) (PIDT-HTBT) doped by D2A1 (PIDT-HTBT-D2A1). The maximum electrical conductivity of PIDT-HTBT-D2A1 is 23.07 S/cm which is comparable to that of FeCl$_3$-doped PIDT-HTBT (Supplementary Table S3) reported in literature[67]. It suggests that efficient molecular doping can be achieved with the coupled reaction doping method and low EA dopants, which overcomes the electron-transfer limitations in the Marcus theory. The results demonstrate that the coupled reaction doping method is a powerful tool that can be widely used for preparing high-performance conducting polymers. The Seebeck coefficient is positive, indicating PIDT-HTBT-D2A1 film is still a p-type material.

In summary, we have demonstrated that the coupled reaction can be used for the efficient molecular doping of conducting polymers with low EA dopants. Adding the additives to react with the intermediate in the doping reaction successfully overcomes the redox potential limitations described by Marcus theory[68], thus leading to the significant improvement of the electrical conductivity of conducting polymers. It can also turn the charge polarity of bipolar polymers from p- to n-type with heavy p-type doping. The coupled reaction doping process shows the potential of wide applications in exploring doping systems to prepare functional conducting polymers, which is a powerful tool for modern organic electronics.

## Methods

### Materials

Tris(pentafluorophenyl)boron, tropylium tetrafluoroborate, 4-acetamido-2, 2, 6, 6-tetramethyl-1-oxopiperidinium tetrafluoroborate and 2, 3, 5, 6-Tetrafluoro-7, 7, 8, 8-tetracyanoquinodimethane were purchased from Bide Pharmatech, Ltd. (Shanghai, China). 2, 2, 6, 6-tetramethyl-1-piperinedinyloxy was purchased from Macklin, China. tris(pentafluorophenyl)boron purity is identified by NMR before use to avoid the influence of impurities such as $H_2O$ (Supplementary Figure S21)[69]. Triethyloxonium tetrafluoroborate, extra dry (water ≤ 10 ppm) acetonitrile, and 1,2-dichlorobenzene were purchased from Energy Chemical, Ltd. (Shanghai, China). Poly (2, 5-bis(2-octyldodecyl)-3, 6-di(thiophen-2-yl)diketopyrrolo[3, 4-c]pyrrole-1, 4-dione-alt-thieno[3, 2-b]thiophen) (DPP4T) and poly (indaceno[1, 2-b:5, 6-b′] dithiophene-4, 7-bis(4-hexylthiophen-2-yl)benzo[c][1, 2, 5] thiadiazole) (PIDT-HTBT) were synthesized from corresponding monomers according to reported literature[70–72] with the monomers purchased from Organtec Ltd. (Beijing, China). 2, 2, 6, 6-Tetramethyl-1-oxopiperidinium tetrafluoroborate and 4-methoxy-2,2,6,6-tetramethyl-1-oxopiperidinium tetrafluoroborate were synthesized from 2, 2, 6, 6-tetramethyl-1-piperinedinyloxy and 4-methoxy-2, 2, 6, 6-Tetramethyl-1-piperinedinyloxy with tetrafluoroboric acid (aqueous solution), 2, 2, 6, 6-tetramethyl-1-oxopiperidinium hexafluorophosphate was synthesized from 2, 2, 6, 6-tetramethyl-1-piperinedinyloxy with hexafluorophosphoric acid(aqueous solution) according to reported literature[73,74]. All commercial reagents were used as received without further purification.

### Film preparation

All solution preparation and film casting were conducted in a nitrogen-filled glovebox with $O_2 < 1$ ppm, and $H_2O < 1$ ppm. Polymer DPP4T and PIDT-HTBT were dissolved in 1, 2-dichlorobenzene ($6.25 \, mg \, mL^{-1}$). Dopant and additive solutions were prepared by dissolving the corresponding small molecules in acetonitrile. Solutions of the polymer and dopant-additive were obtained by mixing the polymer solution with different concentrations of dopant solution and different concentrations of additive solution. Fused rings, such as the DPP unit, were counted as two aromatic rings, the molar ratio of dopant and additive ratio was 1: 1 unless noted. The doped solutions were stirred at room temperature for 2 h before use.

Drop-casting: The obtained mixture was dropped on the clean glass substrates (The glass substrates were cleaned in an ultrasonic bath with a detergent solution, isopropyl alcohol, and acetone, respectively. After the glass substrates were blown dry with $N_2$, they were treated with plasma for 5 min). The films were dried at room temperature in the glovebox for 20 h until all the volatile components were evaporated. Film thicknesses are in the range of 1–2 µm.

**Thermoelectric properties.** The electrical conductivity of the film was measured by commercial equipment NETZSCH SBA-458 (Germany) with a four-probe method. All the measurements were performed under Ar protection at room temperature. A minimum of three to five samples were tested for each data point. The film thickness was measured at KLA-Tencor D120 (America).

**UV-Vis absorption spectroscopy.** Samples for UV−vis−NIR absorbance were prepared by spin-casting pristine or doped conjugated polymer solutions onto clean glass substrates. Analytic PerkinElmer Lambda 950 plus (America) was used to record UV-vis absorption spectra of the films. The spectra were obtained with a scan rate of $20 \, nm \, s^{-1}$ in the range of 400–2000 nm at room temperature. The baseline was collected with a blank glass substrate.

**Electron spin resonance (ESR) measurements.** Samples for EPR measurements were prepared by drop-casting doped polymer solutions in 1,2-dichlorobenzene onto clean PTFE substrates. After drying, the film was cut into thin strips, and loaded into EPR quartz tubes in a nitrogen-filled glovebox ($O_2 < 1$ ppm, $H_2O < 1$ ppm). The measurement was performed on an X-band (9.84 GHz) Burker A300-9.5/12 spectrometer (America) at ambient temperature. The g-value axis was calibrated with a crystalline 2,2-diphenyl-1-picrylhydrazyl (DPPH). The spin-counting was quantified by comparing the doped film intensity of the double integral of the EPR spectra to an external standard curve constructed from TEMPO according to the literature[75]. Parameters for measuring the samples were as follows: microwave frequency 9.87 GHz, modulation frequency 100 kHz, time constant 40.96 ms, conversion time 40 ms, resolution in X 1024, center field 3522 G, modulation amplitude 1.00 G, receiver gain $1 \times 10^2$, sweep width 400 G, and microwave power of 2 mW. EPR peaks were analyzed by using Bruker WinEPR software.

**Hall effect measurements.** The Hall effect measurements were carried out using a physical property measurement system (PPMS - DynaCool, Quantum Design, America), under the FastHall™ model (excitation source current at 100 µA).

**Cyclic voltammetry (CV).** Cyclic voltammetry was performed on a CHI660E instruments in a classic three-electrode cell, with a glassy carbon working electrode, a platinum disc counter electrode, and a silver wire was used as a pseudo-reference electrode. Film measurements were carried out in a 0.1 M tetra-N-butylammonium hexafluorophosphate in acetonitrile. Polymer films were dropped cast on the glassy carbon working electrode from O-dichlorobenzene solutions. The scanning rate was 50 mV/s. Ferrocene/ferrocenium (Fc/Fc⁺) redox pair was used as the internal standard.

**Electrochemical impedance spectroscopy (EIS).** EIS was performed on a CHI660E instruments using a two-point probe approach. An ac voltage of 0.1 V is applied while sweeping the frequency from 1 MHz to 5 Hz.

**Ultra-violet photoelectron spectroscopy (UPS).** UPS and XPS spectrums were measured at Thermo Fisher ESCALAB Xi + . Polymer film was deposited on indium-tin oxide(ITO) glass.

**Grazing-incidence wide-angle X-ray scattering (GIWAXS).** GIWAXS measurements were carried out with Xeuss 3.0 using a 30 W Cu X-ray source (8.05 keV,1.54 Å) and an Eiger2R 1 M detector (distance: 120 mm). The incidence angle is 0.15°. Film samples for GIWAXS measurements were spin-coated on a cleaned Si wafer.

**Electrospray ionization mass spectrometry (ESI-MS).** The DPP4T-D2A1 solutions were further diluted to a concentration below $0.1 \, mg \, ml^{-1}$ before data collection using a Thermo Scientific Q Exactive mass spectrometers under ESI mode.

**Nuclear magnetic resonance (NMR).** Nuclear magnetic resonance (NMR) spectra were measured with a Bruker Avance III HD 400 (America).

### Theoretical calculation

Density functional theory (DFT) calculations were performed with the GAUSSIAN 09 series of programs[76]. DFT method B3LYP with 6−311 + G(d) was used for geometry optimizations, along with Grimme's D3 dispersion correction with BJ damping function. Chlorobenzene as the prototypical solvent implemented in the G09 code was applied by the Self-Consistent Reaction Field (SCRF) method using the polarizable continuum model. Taking into account the calculation efficiency and cost, the 4 repeat units oligomer of DPP4T was calculated. Long alkyl side chains of DPP4T were replaced with methyl,

which would reduce the calculation cost of calculating the electronic structure of the polymer without significantly reducing the calculation accuracy. The adiabatic Ionization potential (IP) and electron affinity (EA) potentials were used to evaluate the ambient stability and the charge injection properties of the DPP4T[77]. The vibrational frequencies at the same level were computed to characterize all optimized structures to evaluate their thermal corrections at 298 K to obtain Gibbs free energies. Please note that the coulomb interaction energy was not involved in the theoretical calculation to save time, which would make the proposed mechanism even more thermodynamically favorable[10,27].

B3LYP/6-311+g(d,p) has an acceptable accuracy, which has been widely used in literatures to calculate the Gibbs free energy of organic semiconductors[7,78]. In this work, Grimme's D3 dispersion correction with BJ damping function was added to improve the accuracy of B3LYP according to literature[79,80]. As suggested by Lu et. al., B3LYP Grimme's D3 dispersion method was a cheap and relatively reliable method, which could provide qualitative calculation results consistent with the accurate method CCDS(T)[81]. The mean absolute deviation of B3LYP Grimme's D3 dispersion method in computed free energy was 10.5 kJ·mol⁻¹, that was 108 meV, as reported by Ho et. al.[82]. The accuracy should be good enough to tell the Gibbs free energy changes in this work which was smaller than $-21.31$ kJ·mol⁻¹, that was $-220$ meV. Therefore, the theoretical calculation indicated that the coupled reaction was thermodynamically favorable.

## Data availability
The data that support the findings of this study are available from the corresponding author upon request. Source data are provided with this paper.

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

## Acknowledgements

H.W. acknowledges financial support from the National Natural Science Foundation of China (Grant No. 52276014 and 51888103), the Fundamental Research Funds for the Central Universities, the World-Class Universities (Disciplines). This work was also supported by the HPC platform and the Instrument Analysis Center, Xi'an Jiaotong University.

## Author contributions

J.P. carried out the experiments; J.W., K.L., and X.D. contributed to sample preparation; Q.L. contributed to the theoretical calculations; D.C., B.C., and J.Y. contributed to the interpretation of the results; H.W. analyzed the data and wrote the manuscript. All authors discussed the results and commented on the manuscript.

## Competing interests

The authors declare no competing interests.
