## [Peer Review File · Nature Communications]

Efficient molecular doping of polymeric semiconductors improved by coupled reactionREVIEWER COMMENTS

Reviewer #1 (Remarks to the Author):

The authors reported a doping method that involves polymeric semiconductor, dopant, and additive, that is a ternary system doping. Additives activate dopants that originally have low activity in electron transfer, as a result, the efficiency of doping is dramatically improved. However, additive activation of dopant molecules is not a very new concept, and is frequently used in the field of organic solar cells. In addition, the dopant D activated by additive A is incorporated into the polymer in the form of AD⁻ (D becomes anion), which likely induces structural disorder. Indeed, this causes monotonic increase of electrical conductivity with respect to dopant molar ratio. Some inconsistencies and poor interpretations are found in the paper. For example, in figure 4(d), the spin density is on the order of 10^{18} cm^{-3} , which is by far below than one charge/monomer unit (10^{21} cm^{-3}). There is also no sufficient explanation regarding the change in polarity of the Seebeck coefficient. Based on the accumulation of internal inconsistency and poor interpretation, I do not think this paper is publishable.

Reviewer #2 (Remarks to the Author):

The manuscript, "Efficient molecular doping of polymeric semiconductors improved by coupled reaction", describes how the addition of two dopants, or what could be viewed as co-dopants, impacts the electrical conductivity of conjugated polymers. Impressively, this doping strategy significantly improves the conductivity of the organic materials to a rather large degree. As noted by the manuscript, it is advantageous in the way the work approached the materials design in that it avoided complicated syntheses for the molecular dopants, which is a different strategy that has been adopted in the community. Of course, this could improve the likelihood of translating the work to practical scenarios. While the overall electrical conductivity of the materials is still on par or lower than champion doped conjugated polymers, this is an interesting design strategy as it may serve as a potential platform for future studies. In this way, the work is likely to be of significant interest to the polymer science and organic electronics communities. How interesting it is to related fields (e.g., those of applied devices) is less obvious given the limits on the ultimate performance

relative to known commercially available materials such as poly(3,4-ethylene dioxythiophene) doped with poly(styrene sulfonate) (PEDOT:PSS) after the PEDOT:PSS has been processed in such a manner that leads to conductivity values on the order of ~ 500 S/cm. The work should also be commended for the detailed explanations provided in both the main text and the Supplemental Information. On the whole, these points were quite clear. Furthermore, on many fronts, the data presented does support the conclusions drawn from the work. However, this is not true in every case.

The following points are situations where the manuscript either does not provide a complete answer or seems to neglect potentially significant lines of question. If these points could be addressed, the manuscript would be more robust and provide a more intriguing effort to the community.

(1) While the body of the paper phrases things more accurately, the abstract is slightly misleading in terms of the potential impact of the work. The abstract could make the reader believe that the manuscript discovered a means by which to circumvent Marcus theory. Clearly, this is not what occurred in the manuscript. It is that the work may have found a clever materials combination that affords a materials system that works efficiently, but in complete agreement with Marcus theory. Revising the abstract to clarify this point is important.

(2) It is not clear why the boron tetrafluoride anion was required for all of the ionic dopants and additives. This was not stated in the manuscript, and it also calls into question some of the universality of the results. That is, are these results specific to this particular salt? If so, what is so special about this anion? Clarification on this point is needed.

(3) The discussions regarding the dopant-to-additive ratios are lacking. This is true as there does not seem to be a completely universal trend on this front. Moreover, this idea directly ties into a question regarding the foundational mechanism proposed in the manuscript (see below). In general, it is not obvious why certain ratios were selected, nor was it obvious why the spacing between the ratios were completed. For instance, in Figure 3a and 3b, there is a huge gap in data between low conductivity (or high Seebeck coefficient) and the next value

where there is a plateau. It is not obvious why this transition regime was not examined in greater detail.

(4) Moreover, it is nice that the electron affinity of the materials was calculated. However, it was not obvious why these were not confirmed with experimental measurements. It is not obvious that the experimental value would match the computational value acquired using Gaussian. This is critical as all of the arguments regarding the mechanism tie back to these ideas. Of course, the manuscript tries to say that computation matched experiment, but this was for a single material, and the match is not great.

(5) One of the biggest overarching concerns of the work is the idea of co-location of the dopant and the additive. All of the arguments and computational data presented necessarily assume that these two species find each other in the thin films, which is a long way from the idealized results of the computation. There is no reason to believe that the single crystal results mentioned by the manuscript would translate to the current study where solution processing and thin film post-processing will greatly impact the final structure. This is a significant gap in terms of connecting the fundamental principles of the work to the end output conductivity.

(6) The second major concern is with respect to the lack of discussion regarding ionic conductivity in this work. That is, it does not appear that there was any effort to decouple the ionic conductivity from the electronic conductivity. In a similar manner, it is not obvious that ionic effects were considered when reporting on the Seebeck coefficient. Of course, this could explain the change in the sign of the Seebeck coefficient in a rather straightforward manner. Thus, it was surprising that the manuscript did not address the presence of ions in the films in a more complete manner.

(7) From an end-use perspective, it was highly surprising that the stability of the thin films was not reported in some manner. As noted by the manuscript itself, doping of conjugated polymers is tricky as doping can lead to long-term stability issues, especially in the presence of oxygen or water. Additionally, from a more fundamental perspective, this would help in terms of the proposed mechanism. There are multiple places where the idea of an anion

residing on the TEMPO unit, or a structure that includes the TEMPO unit, are mentioned. Of course, as noted in the manuscript, it is much easier to oxidize TEMPO than to reduce it, and the reduced version of TEMPO is notoriously unstable as it can be converted to the hydroxyl version quickly. However, the manuscript does not go into detail here. This is a significant weakness.

(8) It was also confusing as to why the DPP4T-D1 thin film did not show a signal in the electron paramagnetic resonance (EPR) spectroscopy data. It should have a radical present as there is a conjugated polymer with a TEMPO dopant present. Does this mean that the DPP4T somehow quenches the TEMPO radical? If so, does this suggest that there are impurities present in the DPP4T that are being removed through the addition of the dopants? Clarity on this point would be much appreciated.

Reviewer #3 (Remarks to the Author):

This manuscript reports an interesting method of doping a polymer semiconductor aided by coupled reaction. In this particular case, p-doping of a hole transport polymer with ionization energy IE is realized by introducing two molecules, each of which having an insufficiently large electron affinity EA to efficiently dope the polymer, but together undergo a chemical reaction which leads to a complex that can accept 1 (or eventually 2) electron from the polymer. Efficient p-doping is achieved.

The method is certainly interesting, as it provides an opportunity to p-dope with molecules that do not have extreme reduction potential, therefore opening the field to a broader choice of materials.

The work presented here is extensive, makes use of several techniques to demonstrate both the mechanism and the efficiency of doping, and of a number of organic compounds to demonstrate the generality of the approach.

Overall, the reviewer believes that this work could be a valuable addition to the field of organic electronics. However, several points need to be addressed before the ms could be considered for publication.

The temperature stability of the doped is not investigated. This is always an important point in organic electronics. The authors are requested to look back at how the doping efficiency as well as the stability of the coupled reaction, and therefore doping, evolves with temperature.

The level of “doping” is very large. Maximum conductivity is achieved for about one D2A1 couple per 2 monomers. This is more alloying than doping. The author avoid talking about carrier mobility and how it is affected by the large concentration of dopant molecules introduced in the polymer. Hole mobility must be measured in these films.

The manuscript reports an interesting shift of the Seebeck coefficient from positive to negative. Unfortunately, the authors do not provide an interpretation. At the large doping concentrations considered here, one could think that the system shifts from the dopants p-doping the dominant polymer at low doping concentration to the polymer n-doping the dominant dopants at large concentration. Transport becomes dominated by electrons hopping between dopants. Any comment on this or other possibilities would be welcome.

In the Introduction, 2nd paragraph, the authors cite reference #9 as an example of mitigating the issue of finding dopant with sufficiently high EA. That is incorrect. That study is about engineering air-stable powerful n-dopants, which otherwise would be readily oxidized upon exposure to ambient.

Page 5, second paragraph: in the discussion on relative EAs of dopants and IE of the polymer, Figure S1 shows the conductivity of DTT4T:A1 reaching $\sim 0.1\text{S/cm.}$, which is several orders of magnitude above the conductivity of undoped DTT4T. Therefore A1 dopes the polymer. How does that square with the fact that $EA(A1) = 3.57\text{ eV}$, more than 1 eV smaller than $IE(DPP4T)$? does this result put some doubt on the theoretical IE and EA values cited here?

Manuscript ID: NCOMMS-24-09903

Title: Efficient molecular doping of polymeric semiconductors improved by coupled reaction

Reviewer Comments to Author:

Reviewer 1

General comments:

The authors reported a doping method that involves polymeric semiconductor, dopant, and additive, that is a ternary system doping. Additives activate dopants that originally have low activity in electron transfer, as a result, the efficiency of doping is dramatically improved. However, additive activation of dopant molecules is not a very new concept and is frequently used in the field of organic solar cells. In addition, the dopant D activated by additive A is incorporated into the polymer in the form of AD⁻ (D becomes anion), which likely induces structural disorder. Indeed, this causes monotonic increase of electrical conductivity with respect to dopant molar ratio. Some inconsistencies and poor interpretations are found in the paper. For example, in figure 4(d), the spin density is on the order of 10^{18} cm^{-3} , which is by far below than one charge/monomer unit (10^{21} cm^{-3}). There is also no sufficient explanation regarding the change in polarity of the Seebeck coefficient. Based on the accumulation of internal inconsistency and poor interpretation, I do not think this paper is publishable.

Authors' response:

Thank you very much for your time and efforts. We sincerely appreciate it.

We agree with the reviewer that the additive activation of dopant molecules in a ternary system doping has been reported in previous works. However, doping mechanisms are far from being well-understood. Exploring new mechanisms for the ternary system doping to improve the doping efficiency is still very interesting to researchers. A few related works have been reported lately (**Nature**, 2023, 622, 287; **Nature**, 2021, 599, 67; **Nature**, 2019, 572, 634; **Adv. Mater.**, 2022, 34, 2102988). For example, Facchetti and Guo *et. al.* reported that the n-doping efficiency of organic semiconductors by the n-type dopant, N-DMBI-H, could be improved by the transition metal nanoparticles additives such as Pt, Au, and Pd, which acted as the catalysts that promoted the doping reaction (**Nature**, 2021, 599, 67). Yamashita *et. al.* reported that the p-doping efficiency of organic semiconductors by the p-type dopant, benzoquinone, could be promoted by ionic liquid additives such as bis(trifluoromethyl sulfonyl) imide (TFSI) and bis(nonafluorobutanesulfonyl) imide (NFSI), which provided counter anions to stable the oxidized organic semiconductor (**Nature**, 2023, 622, 287). Watanabe and Sirringhaus *et. al.* reported that the p-doping efficiency of organic semiconductors by the p-type dopant, 2,3,5,6-tetrafluoro-7,7,8,8-tetracyanoquinodimethane (F4TCNQ), could

be improved by the ionic liquid additives such as Li-TFSI, BMP-TFSI, and TBA-TFSI, *etc.* through a new ion-exchange mechanism (**Nature**, 2019, 572, 634; **Adv. Mater.**, 2022, 34, 2102988). In this work, the p-doping efficiency of organic semiconductors for the p-typed dopant, the nitroxide derivative, was improved by the additives, tris(pentafluorophenyl) borane, through a coupled reaction mechanism. This method has never been reported to improve the doping efficiency of organic semiconductors in previous works, which would avoid the complex synthesis of high electron affinity dopants. This newly reported method would pave a promising way to tune the electrical properties of organic semiconductors for modern organic electronics.

The packing order of the polymer molecules may affect the electrical properties of the films, especially for thin films with the thickness at nanoscales. Typically, the electrical conductivity would be significantly improved in the thin films by packing polymer molecules orderly to increase the carrier mobility (**Proc. Natl. Acad. Sci.**, 2015, 112, 14138; **Chem. Mater.**, 2014, 26, 11, 3471; **Adv. Mater.**, 2012, 24, 2436. PEDOT: PSS >2000 on glass). However, for thick films, it would be hard to pack the polymer molecules very well due to the lack of guidance of the substrates. Therefore, the electrical conductivity of thick films was often lower than that of thin films (**Journal of Electronic Materials**, 2011, 40, 648; **Syn. Met.**, 2010, 160, 2481; **Adv. Mater.**, 2000, 12, 481. PEDOT: PSS <800 free standing). In this work, thick films were prepared by a drop-casting method with a thickness in the range of 1-2 μm . Although the additive **A** might induce structure disorder, the improved doping efficiency still resulted in the improvement of the electrical conductivity of the thick films due to the increased carrier concentration (Hall measurement **Figure S16**). The doping efficiency induced electrical conductivity improvement has also been reported in previous works (**Nature**, 2023, 622, 287; **Nature**, 2021, 599, 67; **Nature**, 2019, 572, 634; **Adv. Mater.**, 2022, 34, 2102988; **Adv. Mater.**, 2020, 32, 2005129). Moreover, it is also likely that the addition of additive **A** increases/maintains the packing order of polymers in the films (**Adv. Mater.** 2024, 2314062). Additional experiment has been performed to show the packing order of the polymer molecules in the drop-casted films before and after the addition of additive **A**. **Figure S10** showed that no obvious difference could be seen in GIWAXS results between the films with and without the addition of the additive **A**.

Simply increasing the dopant molar ratio of the dopants would not cause the increase of the electrical conductivity as shown in **Figure 3a** and **Figure 4a** in the original manuscript. The maximum electrical conductivity for DPP4T-**D2A1**/DPP4T-**D1A1** was obtained when the dopant molar ratio of **D2A1/D1A1** was ~0.3, which was much higher than that for DPP4T-**D2**/DPP4T-**D1** with the dopant molar ratio of **D2/D1**

of even >1.3 . The results demonstrated that a higher dopant molar ratio of **D2/D1** did not mean a higher electrical conductivity in DPP4T-**D2**/DPP4T-**D1** films. Additional experiments have been performed by adding the commonly used dopant F4TCNQ as an additive. **Figure S3** showed that the maximum electrical conductivity of DPP4T-**D2F4TCNQ** was much lower than that of DPP4T-**D2A1**. It suggested that the additive A in this work played an important role, which promoted the p-doping of the organic semiconductors. A coupled reaction mechanism was proposed with the support of the experimental and theoretical results.

Figure S3. The electrical conductivity of DPP4T-**D2F4TCNQ** varies with dopant molar ratio.

It was known that the ESR/EPR often underestimated the carrier density at high doping levels because Pauli-type magnetic susceptibility was often observed and the spin density corresponded only to the subset of spins in close vicinity to the Fermi level. (**J. Am. Chem. Soc.**, 2022, 144, 3005; **Nature** 2019, 572, 634; **Nat. Mater.**, 2016, 15, 896; **Sci. Adv.**, 2020, 6, eaay8065). As stated in the original manuscript, the spin density was used to qualitatively understand the doping level of the polymers, "The spin concentration has been calculated from the double integral of the signal as shown in **Figure 4d** and **Figure S13**, which can be used to qualitatively understand the doping level of polymers". The trend of the doping level of polymers obtained with ESR results was consistent with that obtained with UV-vis spectra and Hall measurements. The quantitative analysis of the carrier density was performed according to the Hall measurements. Although it was still rough, the results were much closer to the real carrier density of the doped polymer films than the ESR results.

At last, the polarity change of the Seebeck coefficient is unusual for heavily p-/n-doped polymers. This novel property was reported in only a few works recently (**Nat. Mater.**, 2021, 20, 518; **Adv. Mater.**, 2022, 34, 2106624; **CCS Chem.**, 2021, 3, 2482; **Adv. Mater.**, 2018, 30, 1802850; **Adv. Mater.**, 2018, 30, 1804290; **Phys. Chem. Chem. Phys.**, 2016, 18, 29199). The polarity change should be attributed to the high doping

level (shown in **Figure 4d** and **Figure S16**), which might move the Fermi level below the valence band of the polymeric semiconductor as suggested in previous literature. (**Adv. Mater.**, 2022, 34, 2106624; **CCS Chem.**, 2021, 3, 2482; **Adv. Mater.**, 2018, 30, 1802850; **Adv. Mater.**, 2018, 30, 1804290; **Phys. Chem. Chem. Phys.**, 2016, 18, 29199). The polarity changes of Seebeck coefficient from p- to n- might also be due to the electrons delocalized from the crystalline regions as suggested by Graham *et. al.* (**Nat. Mater.**, 2021, 20, 518). However, the mechanism of the polarity change of the Seebeck coefficient is still not very convincing because the doping mechanism is far away from being well understood which may be affected by a lot of factors such as the packing order of polymer molecules, polymer structure, morphology and dopant distribution, the size and structure of the dopant, *etc.* More discussion has been provided in the revised manuscript. We hope the reviewer could understand that the complement study of the polarity change of the Seebeck coefficient would be out of the research slope of this work as well. Because the main-focus of the work was to report a new coupled reaction induced p-type doping, which significantly improved the electrical conductivity of polymers without increasing the electron affinity of dopants. These relatively low electron affinity dopants could also lead to the polarity change of the Seebeck coefficient like the high electron affinity dopant like FeCl₃, which might be helpful to clearly understand the unusual doping behavior in organic semiconductors in the future.

Related contents have been added in the revised manuscript.

Reviewer 2

General comments:

The manuscript, “Efficient molecular doping of polymeric semiconductors improved by coupled reaction”, describes how the addition of two dopants, or what could be viewed as co-dopants, impacts the electrical conductivity of conjugated polymers. Impressively, this doping strategy significantly improves the conductivity of the organic materials to a rather large degree. As noted by the manuscript, it is advantageous in the way the work approached the materials design in that it avoided complicated syntheses for the molecular dopants, which is a different strategy that has been adopted in the community. Of course, this could improve the likelihood of translating the work to practical scenarios. While the overall electrical conductivity of the materials is still on par or lower than champion doped conjugated polymers, this is an interesting design strategy as it may serve as a potential platform for future studies. In this way, the work is likely to be of significant interest to the polymer science and organic electronics communities. How interesting it is to related fields (e.g., those of applied devices) is less obvious given the limits on the ultimate performance relative to known commercially available materials such as poly(3,4-ethylene dioxythiophene) doped with poly(styrene sulfonate) (PEDOT:PSS) after the PEDOT:PSS has been processed in such a manner that leads to conductivity values on the order of ~500 S/cm. The work should also be commended for the detailed explanations provided in both the main text and the Supplemental Information. On the whole, these points were quite clear. Furthermore, on many fronts, the data presented does support the conclusions drawn from the work. However, this is not true in every case.

The following points are situations where the manuscript either does not provide a complete answer or seems to neglect potentially significant lines of question. If these points could be addressed, the manuscript would be more robust and provide a more intriguing effort to the community.

Authors' response:

Thank you very much for your time and efforts. We sincerely appreciate it.

Chemical doping is an important way to tune the electrical properties of polymeric semiconductors for modern organic electronics. However, the doping mechanisms are far from being well-understood. Exploring new doping approaches to improve doping efficiency is still very interesting to researchers (**Nature**, 2023, 622, 287; **Nature**, 2021, 599, 67; **Nature**, 2019, 572, 634; **Adv. Mater.**, 2022, 34, 2102988). We reported a new doping method that was named the coupled reaction doping, which could offer efficient molecular doping of conducting polymer when the EA of a dopant was lower than the IP of a polymer. The newly developed doping method incompletely agreed with traditional chemical doping that generally obeyed the Marcus theory. Furthermore, the doped polymer exhibited n-type properties after heavily p-doping. It was like the polymers

doped by high EA dopant FeCl₃ (**Nat. Mater.**, 2021, 20, 518; **Adv. Mater.**, 2022, 34, 2106624; **CCS Chem.**, 2021, 3, 2482; **Adv. Mater.**, 2018, 30, 1802850; **Adv. Mater.**, 2018, 30, 1804290; **Phys. Chem. Chem. Phys.**, 2016, 18, 29199), which might be helpful to understand the recently reported unusual behavior of organic semiconductors in future. Although the electric conductivity of ~40 S/cm for the drop-casted thick films in micrometer scale was lower than that of ~500 S/cm for the state-of-the-art p-type polymer PEDOT:PSS, it was of significant interest to serve as a potential high electrical conductive material for future studies because the electrical conductivity was already higher than some of the values reported in previous works (<10⁻³ S/cm, **Nat. Mater.**, 2017, 16, 1209; <10⁻³ S/cm, **Nat. Mater.**, 2019, 18, 1327; ~33 S/cm, **Adv. Energy Mater.**, 2020, 10, 2002521). According to the literature, the electrical conductivity of polymeric films was affected by many factors, which was very likely to be improved after optimizing the preparation conditions, such as making thin films, *etc.* (**Adv. Mater.**, 2012, 24, 2436; **Chem. Mater.**, 2014, 26, 3471; **Adv. Mater.**, 2014, 26, 2268; **Adv. Mater.**, 2015, 27, 2317). It was noticed that the electrical conductivity of PEDOT:PSS films was also not that high (~1 S/cm) in the early reported works (**Energy Environ. Sci.**, 2012, 5, 9662).

Related contents have been added in the revised manuscript. We hope this clarification addresses your concerns regarding the electrical conductivity of the DPP4T films. We appreciate your time and consideration, and we remain open to any further suggestions or feedback you may have.

Reviewer's Comments 1:

While the body of the paper phrases things more accurately, the abstract is slightly misleading in terms of the potential impact of the work. The abstract could make the reader believe that the manuscript discovered a means by which to circumvent Marcus theory. Clearly, this is not what occurred in the manuscript. It is that the work may have found a clever materials combination that affords a materials system that works efficiently, but incomplete agreement with Marcus theory. Revising the abstract to clarify this point is important.

Authors' response:

Thank you very much for your good suggestion. The abstract has been replaced with the following one:

“Chemical doping of conducting polymers is an important way to tune the electrical properties of polymeric semiconductors for modern organic electronics. The traditional doping process is determined by the redox potential, which generally requires the electron affinity (EA) of the dopant to match with the ionic potential (IP) of the polymer for p-type doping. Here, we report a different doping process named the coupled reaction

doping process, which can offer efficient molecular doping of conducting polymer when the EA of a dopant is lower than the IP of a polymer. During the doping process, the chemical reaction between the dopant and the polymer is promoted by introducing a thermodynamically favorable reaction via adding additives that are highly reactive to the reduction product of the dopant to form a coupled reaction, thus allowing the electrons to transfer from the polymer to the dopant. This coupled reaction doping process overcomes the redox potential limitations in Marcus theory, which results in high doping levels up to one charge per repeated unit for conducting polymers with low EA dopants, thus significantly improving the electrical conductivity of polymers. It can also turn the charge polarity of polymer from p- to n-type with heavy p-type doping. This coupled reaction doping process shows the potential of wide applications in exploring new doping systems to prepare functional conducting polymers, which could be a powerful tool for modern organic electronics.”

Reviewer's Comments 2:

It is not clear why the boron tetrafluoride anion was required for all of the ionic dopants and additives. This was not stated in the manuscript, and it also calls into question some of the universality of the results. That is, are these results specific to this particular salt? If so, what is so special about this anion? Clarification on this point is needed.

Authors' response:

Thank you very much for your time and efforts. We sincerely appreciate it. The boron tetrafluoride anion is not indispensable.

Additional experiments have been performed to check if the boron tetrafluoride anion is indispensable in the coupled reaction induced p-type doping process. The dopant TEMPO⁺PF₆⁻ (**D5**) was synthesized according to the literature (**Chem. Commun.**, 2019, 55, 6536; **J. Org. Chem.**, 2008, 73, 4750). **Figure S5a** showed that significant improvement could also be obtained while doping DPP4T with **D5A1** (DPP4T-**D5A1**) with a maximum electrical conductivity of 34.5 S/cm and the polarity of Seebeck coefficient also changed from p- to n-type after increasing the dopant molar ratio up to >0.5 (**Figure S5b**). The results were similar to that of DPP4T-**D2A1**, which demonstrated the wide compatibility of the coupled reaction doping process.

Figure S5. (a) The conductivity of DPP4T-D5A1 varies with dopant molar ratio; (b) The Seebeck coefficient of DPP4T-D5A1 varies with dopant molar ratio.

Related contents have been added in the revised manuscript.

Reviewer's Comments 3:

The discussions regarding the dopant-to-additive ratios are lacking. This is true as there does not seem to be a completely universal trend on this front. Moreover, this idea directly ties into a question regarding the foundational mechanism proposed in the manuscript (see below). In general, it is not obvious why certain ratios were selected, nor was it obvious why the spacing between the ratios were completed. For instance, in Figure 3a and 3b, there is a huge gap in data between low conductivity (or high Seebeck coefficient) and the next value where there is a plateau. It is not obvious why this transition regime was not examined in greater detail.

Authors' response:

Thank you very much for your time and efforts. We sincerely appreciate it. The ratio and the spacing between the ratios appeared casual. More points have been added in **Figure 3a** and **3b**.

We started doping DPP4T with **D1(D2)** and **A1** at the dopant-to-additive ratio of 1:1 because literature indicated that the formation of **D1⁻** and **A1** at the ratio of 1:1 (**J. Am. Chem. Soc.**, 2017, 139, 10062). Then, the electrical conductivity as a function of the dopant molar ratio was tested as shown in **Figure 3a** of the original manuscript. Dopant molar ratios of 0.1, 0.3, 0.5, 0.7, 0.9, 1.1, 1.3 were used casually. It was found that a significant improvement in terms of the electrical conductivity was observed at the dopant molar ratio of 0.3 in both DPP4T-**D1A1** and DPP4T-**D2A1** films. After that, the dopant-to-additive ratio was then screened with the fixed **D2** molar ratio of 0.3 for DPP4T-**D2A1** films as shown in **Figure S6** of the supporting information. Because the electrical conductivity of DPP4T-**D2A1** film was higher than that of DPP4T-**D1A1** film. The **A1** molar ratio of 0.1, 0.3, 0.5, 0.7, 0.9, 1.1, 1.3 were used casually. The results indicated that the maximum electrical conductivity was obtained when the **A1** molar ratio

equaled to the **D2** molar ratio (dopant-to-additive ratio 1:1).

Additional experiments have been performed by doping DPP4T with **D1A1** at the low dopant molar ratios. The electrical conductivities were 0.01 and 0.07 S/cm for DPP4T-**D1A1** at the dopant molar ratio of 0.01 and 0.05, respectively. However, the Seebeck coefficient was not measurable when the dopant molar ratio of **D1A1** was <0.1. The electrical conductivity and the Seebeck coefficient of DPP4T-**D1A1** at the dopant molar ratio of 0.2 were 3.4 S/cm and 156 $\mu\text{V/K}$, respectively. (**Figure 3b**).

Related contents have been added in the revised manuscript.

Figure 3. The electric conductivity (a) and the Seebeck coefficient (b) of DPP4T-D1A1 as a function of the dopant molar ratio.

Related contents have been added in the revised manuscript.

Reviewer's Comments 4:

Moreover, it is nice that the electron affinity of the materials was calculated. However, it was not obvious why these were not confirmed with experimental measurements. It is not obvious that the experimental value would match the computational value acquired using Gaussian. This is critical as all of the arguments regarding the mechanism tie back to these ideas. Of course, the manuscript tries to say that computation matched experiment, but this was for a single material, and the match is not great.

Authors' response:

Thank you very much for your great suggestions. In general, EA and IP values can be used to predict whether doping reactions occur. For P-type doping, the EA value of the dopant is often higher than the IP value of the polymer to ensure an efficient molecular doping (*Chem. Soc. Rev.*, 2020, 49, 7210; *J. Phys. Chem. Lett.*, 2020, 11, 3928). However, in this work, the EA value of the dopants is lower than the IP value of the polymer, which still leads to efficient p-type doping due to the coupled reaction.

Additional experiments were performed to evaluate the EA and IP of the dopants

and polymers with an electrochemical method according to the following equation (**Chem. Eng. J.**, 2022, 450, 138135).

$$E_{\text{HOMO}} = -\text{IP} = -e(\varphi_{\text{ox}} - \varphi_{\text{Fc}^+/\text{Fc}} + 4.8) \text{ (eV)}$$

$$E_{\text{LUMO}} = -\text{EA} = -e(\varphi_{\text{red}} - \varphi_{\text{Fc}^+/\text{Fc}} + 4.8) \text{ (eV)}$$

The obtained experimental IP and EA values were shown in **Table S1**, which matched well with the theoretical values acquired using Gaussian for DPP4T, **A1**(B(C₆F₅)₃), **D1**(TEMPO). In addition, the IP value of DPP4T was further evaluated by ultraviolet photoelectron spectroscopy (UPS) which also agreed well with the theoretical IP value (**Table S1**). Similar IP and EA values were reported in previous literature for DPP4T (IP: 4.97-5.2 eV, **Nat. Mat.**, 2021, 20, 518; **J. Am. Chem. Soc.**, 2015, 137, 1314; **J. Am. Chem. Soc.**, 2011, 133,15073; **J. Mater. Chem. C.**, 2013, 1, 4423), **A1**(EA: 3.03-3.31 eV, **Angew. Chem. Int. Ed.**, 2020, 59, 22210), **D1**(2.33 eV, **J. Org. Chem.**, 2008,73, 6763) .

Table S1. Comparison of IP/EA calculations values and the experimental values obtained with electrochemical method

	IP/EA value (eV)	
	calculate	experimental
DPP4T	IP:4.69	IP: 4.89 (4.64 UPS value)
B(C ₆ F ₅) ₃ (D1)	EA: 3.57	EA: 3.66
TEMPO(A1)	EA: 2.16	EA: 2.66

Figure S1. Cyclic voltammetry curve of (a) Cp_2Fe , (b) DPP4T, (c) TEMPO, and (d) $\text{B}(\text{C}_6\text{F}_5)_3$; UPS spectrum of Au(e) and DPP4T(f).

Related contents have been added in the revised manuscript.

Reviewer's Comments 5:

One of the biggest overarching concerns of the work is the idea of co-location of the dopant and the additive. All of the arguments and computational data presented necessarily assume that these two species find each other in the thin films, which is a long way from the idealized results of the computation. There is no reason to believe that the single crystal results mentioned in the manuscript would translate to the current study where solution processing and thin film post-processing will greatly impact the final structure. This is a significant gap in terms of connecting the fundamental principles of the work to the end output conductivity.

Authors' response:

Thank you very much for your good question. We sincerely appreciate your time and

efforts.

The idea of the formation of the coordination compound between the dopant and the additive was based on the following reasons: Firstly, the dopant **D2** and the additive **A1** were surrounded by each other since they were mixed thoroughly with the polymer in solution during the film preparation process as described in the original supporting information. "Solutions of the polymer and dopant-additive were obtained by mixing the polymer solution with different concentrations of dopant solution and different concentrations of additive solution." Secondly, the reduction products of **D2** ions thermodynamically preferred to form a coordination compound with the Lewis acid **A1** according to the theoretical calculation and the experimental results reported in previous works (*J. Am. Chem. Soc.*, 2012, 134, 19350; *Inorg. Chem.*, 2014, 53, 11377; *Chem. Eur. J.*, 2016, 22, 9504). The formation of the single crystal of **D2·A1** in a complex solution system reported in previous works (*J. Am. Chem. Soc.*, 2017, 139, 10062) strongly demonstrated the good coordination capacity between **D2** and **A1**. At last, the electrospray ionization mass spectrometry (ESI-MS) results were shown in our original manuscript, which showed the existence of the molecular ion peak at $m/z=668$ for **D2·A1** in **Figure S17** of the original supporting information. Based on the above facts, it is reasonable to believe that the coordination compound of the dopant and the additive forms in during the film preparation process which subsequently induces the efficient molecular doping of the polymer.

Related contents have been added in the revised manuscript.

Reviewer's Comments 6:

The second major concern is with respect to the lack of discussion regarding ionic conductivity in this work. That is, it does not appear that there was any effort to decouple the ionic conductivity from the electronic conductivity. In a similar manner, it is not obvious that ionic effects were considered when reporting on the Seebeck coefficient. Of course, this could explain the change in the sign of the Seebeck coefficient in a rather straightforward manner. Thus, it was surprising that the manuscript did not address the presence of ions in the films in a more complete manner.

Authors' response:

Thank you very much for your great suggestions. We sincerely appreciate it. The materials are electronic conduction dominated materials rather than ionic conduction dominated materials.

For the DPP4T-**D2A1** films, they all had a high electrical conductivity of > 2.7 S/cm, which indicated that they should be electronic conduction dominated materials. Because the ions transport was slow and ionic conduction dominated materials typically had a

poor electrical conductivity of <0.5 S/cm (**Adv. Energy Mater.**, 2015, 5, 1500044). Additional experiments were performed to demonstrate the dominate electronic conduction in DPP4T-D2A1 films with AC impedance spectroscopy. A detailed analysis of the electrical behavior of the films was carried out by a two-probe AC method. The Nyquist plots of DPP4T-D2A1 films at the dopant molar ratio of 0.3 (p-type) and 1.3 (n-type) showed data regrouped around a point (**Figure S18a**, like for an ideal resistance), indicating a dominant electronic conduction of these films (**Adv. Energy Mater.**, 2015, 5, 1500044; **Electrochem. Solid ST.**, 2007, 10, A65). **Figure S18b** also showed that the normalized electrical conductivities for the films were constant up to 10^4 and 10^3 Hz for the dopant molar ratio of 0.3 (p-type) and 1.3 (n-type), respectively. Indicating that electronic transport was dominated over ionic effects since the increase of the electrical conductivity often occurred at a very low frequency (about 0.3 Hz, **J. Mater. Chem. A**, 2014, 2, 20676; **Adv. Mater.**, 2016, 28, 9545; **Adv. Mater.**, 2022, 34, 2106624). Therefore, the ionic effects on the Seebeck coefficient and the electrical conductivity in the films may be ignored.

Figure S18. The intercept of the Nyquist plots horizontal axis at 0.3(a) and 1.3(b) dopant molar ratio; The change of conductivity at different frequencies of DPP4T-D2A1 at 0.3(c) and 1.3(d) dopant molar ratio.

The polarity change of the Seebeck coefficient is unusual for heavily p-/n-doped polymers. This novel property was reported in only a few works recently (**Nat. Mater.**, 2021, 20, 518; **Adv. Mater.**, 2022, 34, 2106624; **CCS Chem.**, 2021, 3, 2482; **Adv.**

Mater., 2018, 30, 1802850; **Adv. Mater.**, 2018, 30, 1804290; **Phys. Chem. Chem. Phys.**, 2016, 18, 29199). The polarity change should be attributed to the high doping level (shown in **Figure 4d** and **Figure S16**), which might move the Fermi level below the valence band of the polymeric semiconductor as suggested in previous literature. (**Adv. Mater.**, 2022, 34, 2106624; **CCS Chem.**, 2021, 3, 2482; **Adv. Mater.**, 2018, 30, 1802850; **Adv. Mater.**, 2018, 30, 1804290; **Phys. Chem. Chem. Phys.**, 2016, 18, 29199). The polarity changes of Seebeck coefficient from p- to n- might also be due to the electrons delocalized from the crystalline regions as suggested by Graham *et. al.* (**Nat. Mater.**, 2021, 20, 518). However, the mechanism of the polarity change of the Seebeck coefficient is still not very convincing because the doping mechanism is far away from being well understood which may be affected by a lot of factors such as the packing order of polymer molecules, polymer structure, morphology and dopant distribution, the size and structure of the dopant, *etc.* More discussion has been provided in the revised manuscript. We hope the reviewer could understand that the complement study of the polarity change of the Seebeck coefficient would be out of the research slope of this work as well. Because the main-focus of the work was to report a new coupled reaction induced p-type doping, which significantly improved the electrical conductivity of polymers without increasing the electron affinity of dopants. These relatively low electron affinity dopants could also lead to the polarity change of the Seebeck coefficient like the high electron affinity dopant like FeCl₃, which might be helpful to clearly understand the unusual doping behavior in organic semiconductors in the future.

Related contents have been added in the revised manuscript.

Reviewer's Comments 7:

From an end-use perspective, it was highly surprising that the stability of the thin films was not reported in some manner. As noted by the manuscript itself, doping of conjugated polymers is tricky as doping can lead to long-term stability issues, especially in the presence of oxygen or water. Additionally, from a more fundamental perspective, this would help in terms of the proposed mechanism. There are multiple places where the idea of an anion residing on the TEMPO unit, or a structure that includes the TEMPO unit, are mentioned. Of course, as noted in the manuscript, it is much easier to oxidize TEMPO than to reduce it, and the reduced version of TEMPO is notoriously unstable as it can be converted to the hydroxyl version quickly. However, the manuscript does not go into detail here. This is a significant weakness.

Authors' response:

Thank you very much for your good suggestions. Additional experiments have been performed to identify the stability of the doped films.

We agree with the reviewer that the dopant TEMPO and related reduced version are

not stable. Additional experiments also showed that the electrical conductivity of DPP4T-D2A1 at 0.3 dopant molar ratio dropped fast while being kept in the ambient at room temperature, which became less than half of the initial value within 24 h. However, these films exhibited a good stability while they were laminated by polyethylene terephthalate (PET) films. **Figure S7** showed that the electrical conductivity of DPP4T-D2A1 film at 0.3 dopant molar ratio could be maintained >85% for 7 days while being laminated by PET films, which indicated a potential way to keep the electrical conductivity of the film for possible practical applications in the future.

Figure S7. The electrical conductivity of laminated DPP4T-D2A1 at 0.3 dopant molar ratio as a function of time.

Related contents have been added in the revised manuscript.

Reviewer's Comments 8:

It was also confusing as to why the DPP4T-D1 thin film did not show a signal in the electron paramagnetic resonance (EPR) spectroscopy data. It should have a radical present as there is a conjugated polymer with a TEMPO dopant present. Does this mean that the DPP4T somehow quenches the TEMPO radical? If so, does this suggest that there are impurities present in the DPP4T that are being removed through the addition of the dopants? Clarity on this point would be much appreciated.

Authors' response:

Thank you very much for your great suggestions. It is likely that the TEMPO has been evaporated which leads to the disappear of the signal of the TEMPO radical.

Additional experiments were performed to test the EPR of DPP4T-D1 and D1 only in both solution and dry status. **Figure S14** showed that both DPP4T-D1 and D1 only in solutions had strong signals which were assigned to the TEMPO radical (**Free Radical Biology and Medicine**, 2014, 77, 64; **Photochem. Photobiol.**, 2012, 88, 1442; **Nature** 1976, 263, 442), while DPP4T-D1 and D1 only dry films showed no signal. The results

demonstrated that DPP4T would not quench TEMPO radical. It was hypothesized that the TEMPO had been evaporated in the dry films since the boiling point of TEMPO (193 °C) was close to that of the used solvent (179 °C) (boiling point data obtained from www.chemspider.com of The Royal Society of Chemistry).

Figure S14. (a) EPR signal of DPP4T-D1 in solution and dry film; (b) pure TEMPO in solution and dry on a substrate.

Related contents have been added in the revised manuscript.

Reviewer 3

General comments:

This manuscript reports an interesting method of doping a polymer semiconductor aided by coupled reaction. In this particular case, p-doping of a hole transport polymer with ionization energy IE is realized by introducing two molecules, each of which having an insufficiently large electron affinity EA to efficiently dope the polymer, but together undergo a chemical reaction which leads to a complex that can accept 1 (or eventually 2) electron from the polymer. Efficient p-doping is achieved. The method is certainly interesting, as it provides an opportunity to p-dope with molecules that do not have extreme reduction potential, therefore opening the field to a broader choice of materials. The work presented here is extensive, makes use of several techniques to demonstrate both the mechanism and the efficiency of doping, and of a number of organic compounds to demonstrate the generality of the approach. Overall, the reviewer believes that this work could be a valuable addition to the field of organic electronics. However, several points need to be addressed before the ms could be considered for publication.

Authors' response:

Thank you very much for your time and efforts. We sincerely appreciate it.

Reviewer's Comments 1:

The temperature stability of the doped is not investigated. This is always an important point in organic electronics. The authors are requested to look back at how the doping efficiency as well as the stability of the coupled reaction, and therefore doping, evolves with temperature.

Authors' response:

Thank you very much for your suggestions. We sincerely appreciate it.

Additional experiments were performed to show the temperature stability of the DPP4T-D2A1 films. **Figure S8a** showed the temperature dependent electrical conductivity of DPP4T-D2A1 films at the temperature range of 25 °C-120 °C. It indicated that the electrical conductivity maintained when the temperature was lower than 75 °C. After that the electrical conductivity of DPP4T-D2A1 films decreased, which was only 11% at the 120 °C. **Figure S8b** showed that the electrical conductivities of the DPP4T-D2A1 films prepared at different temperature. It revealed that the electrical conductivity decreased with the increase of the temperature since TEMPO and related reduction compounds were generally unstable as reported in previous works (**Angew. Chem. Int. Ed.**, 2011, 50, 5034; **Synthesis**, 1996, 10, 1153; **Adv. Synth. Catal.**, 2004, 346, 1051; **New J. Chem.**, 2005, 29, 1308).

Figure S8 (a) Temperature dependent electrical conductivity of DPP4T-D2A1 films at the temperature range of 25°C-120°C.; (b) The electrical conductivities of DPP4T-D2A1 films prepared at different temperatures.

Reviewer's Comments 2:

The level of “doping” is very large. Maximum conductivity is achieved for about one D2A1 couple per 2 monomers. This is more alloying than doping. The author avoid talking about carrier mobility and how it is affected by the large concentration of dopant molecules introduced in the polymer. Hole mobility must be measured in these films.

Authors' response:

Thank you very much for your good suggestion.

We agree with the reviewer that the dopant molar ratio is high for DPP4T-D2A1. However, it is still called “doping” for polymeric materials since there is no evidence that the D2A1 forms a coordination compound with the polymer (**Nat. Mater.**, 2021, 20, 518).

Additional experiments have been performed to show the carrier mobility change as a function of the dopant molar ratio. **Figure S15a** showed that the carrier mobility of DPP4T-D2A1 film was in the range of 0.06-2.5 cm²/(V·s), which decreased with the increase of the dopant molar ratio since the dopants were non-conductive materials. Then, it increased at heavily doping level with the dopant molar ratio of 0.9. The rise of the carrier mobility should be due to formation of the crystalline regions as suggested by Graham *et. al.* (**Nat. Mater.**, 2021, 20, 518). **Figure S15b** showed that the electrical conductivity of DPP4T-D2A1 increased with the decrease of the carrier mobility, which was also commonly reported in previous works since the electrical conductivity of polymeric materials was affected by both the carrier mobility and the carrier concentration that was related to the efficient molecular doping level (**Phy. Rev. Lett.**, 2011 107, 066601; **Phy. Rev. B**, 2012 85, 035313; **Phys. Rev. Lett.**, 2004, 93, 086602; **Sci. Rep.**, 2016, 6, 23650; **Adv. Funct. Mater.**, 2020, 30, 1903617).

Figure S15. (a) Hall mobility μ_{Hall} of DPP4T-D2A1 as a function of the different dopant molar ratio; (b) Hall mobility versus the electrical conductivity.

Reviewer's Comments 3:

The manuscript reports an interesting shift of the Seebeck coefficient from positive to negative. Unfortunately, the authors do not provide an interpretation. At the large doping concentrations considered here, one could think that the system shifts from the dopants p-doping the dominant polymer at low doping concentration to the polymer n-doping the dominant dopants at large concentration. Transport becomes dominated by electrons hopping between dopants. Any comment on this or other possibilities would be welcome.

Authors' response:

Thank you very much for your suggestion. Tuning the concentration of one component to change the polarity may be observed in the all-polymer donor-acceptor heterojunctions because of the ground-state electron transfer as reported by Berggren and Fabiano *et. al.* (**Nat. Mater.**, 2020, 19, 738). However, it is unlikely to be happened in work as the dopants are not semiconductors which are not possible to show high electrical conductivity up to 16.2 S/cm without crystalline structures for DPP4T-D2A1 at the dopant molar ratio of 0.7 (Seebeck coefficient -8.8 $\mu\text{V/K}$) as shown in **Figure 4a** in the original manuscript.

The polarity change of the Seebeck coefficient is unusual for heavily p-/n-doped polymers. This novel property was reported in only a few works recently (**Nat. Mater.**, 2021, 20, 518; **Adv. Mater.**, 2022, 34, 2106624; **CCS Chem.**, 2021, 3, 2482; **Adv. Mater.**, 2018, 30, 1802850; **Adv. Mater.**, 2018, 30, 1804290; **Phys. Chem. Chem. Phys.**, 2016, 18, 29199). The polarity change should be attributed to the high doping level (shown in **Figure 4d** and **Figure S16**), which might move the Fermi level below the valence band of the polymeric semiconductor as suggested in previous literature. (**Adv. Mater.**, 2022, 34, 2106624; **CCS Chem.**, 2021, 3, 2482; **Adv. Mater.**, 2018, 30, 1802850; **Adv. Mater.**, 2018, 30, 1804290; **Phys. Chem. Chem. Phys.**, 2016, 18, 29199). The polarity changes of Seebeck coefficient from p- to n- might also be due to the electrons

delocalized from the crystalline regions as suggested by Graham *et. al.* (**Nat. Mater.**, 2021, 20, 518). However, the mechanism of the polarity change of the Seebeck coefficient is still not very convincing because the doping mechanism is far away from being well understood which may be affected by a lot of factors such as the packing order of polymer molecules, polymer structure, morphology and dopant distribution, the size and structure of the dopant, *etc.* More discussion has been provided in the revised manuscript. We hope the reviewer could understand that the complement study of the polarity change of the Seebeck coefficient would be out of the research slope of this work as well. Because the main-focus of the work was to report a new coupled reaction induced p-type doping, which significantly improved the electrical conductivity of polymers without increasing the electron affinity of dopants. These relatively low electron affinity dopants could also lead to the polarity change of the Seebeck coefficient like the high electron affinity dopant like FeCl₃, which might be helpful to clearly understand the unusual doping behavior in organic semiconductors in the future. Further work is still undergoing to fully understand the polarity changes.

Related contents have been added in the revised manuscript.

Reviewer's Comments 4:

In the Introduction, 2nd paragraph, the authors cite reference #9 as an example of mitigating the issue of finding dopant with sufficiently high EA. That is incorrect. That study is about engineering air-stable powerful n-dopants, which otherwise would be readily oxidized upon exposure to ambient.

Authors' response:

Thank you very much for pointing this out. We are sorry for the mistake. Reference #9 has been removed in the revised manuscript.

Reviewer's Comments 5:

Page 5, second paragraph: in the discussion on relative EAs of dopants and IE of the polymer, Figure S1 shows the conductivity of DTT4T:A1 reaching ~0.1S/cm., which is several orders of magnitude above the conductivity of undoped DTT4T. Therefore A1 dopes the polymer. How does that square with the fact that EA(A1) = 3.57 eV, more than 1 eV smaller than IE(DPP4T)? does this result put some doubt on the theoretical IE and EA values cited here?

Authors' response:

Thank you very much for your time and efforts. We sincerely appreciate it. The theoretical IE and EA values were close to the values reported in the previous works, which should be correct. For the electrical conductivity of 0.1 S/cm of DPP4T-A1, the doping process was complex and could be affected by many factors. In the meanwhile,

the slight doping of DPP4T by **A1** would not harm the major conclusion that the coupled reaction formed by two weak dopants (dopant and additive) could promote the doping efficiency of DPP4T-**A1** since the doping process is un-sufficient.

The electron affinity (EA) shows the ability of the dopants to accept an electron and the ionization potential (IP) shows the ability of the polymer to donate an electron. For p-type doping, the EA of the dopant is generally required to match or exceed the IP of polymeric semiconductors to ensure the complete transferring an electron from the polymeric semiconductor DPP4T to the dopant A1 (**J. Phys. Chem. Lett.**, 2020, 11, 3928; **Chem. Soc. Rev.**, 2020, 49, 7210).

The EA of 3.57 eV for **A1** and the IP of 4.69 eV for the polymer DPP4T were calculated by using Gaussian 09 at the level of the B3LYP hybrid functional. The obtained values were close to literature values (EA for A1, 3.03-3.31 eV, **Angew. Chem. Int. Ed.**, 2020, 59, 22210; IP for DPP4T, 4.97-5.2 eV, **J. Am. Chem. Soc.**, 2015, 137, 1314; **J. Am. Chem. Soc.**, 2011, 133, 15073–15084; **J. Mater. Chem. C**, 2013, 1, 4423). In principle, it was unfavorable for the electrons to transfer from DPP4T to **A1** (p-doping). However, the doping process was much more complex than that it was expected, which would be affected by many factors, such as the packing order of polymer molecules, polymer structure, morphology, dopant distribution, the size and structure of the dopant, *etc* (**Nat. Mater.**, 2021, 20, 518). There were several potential reasons for the low electrical conductivity of ~0.1 S/cm in the DPP4T-**A1** films. Firstly, the EA of 3.57 eV for A1 represented that the maximum possibility of an electron could be accepted by **A1**. When mixing **A1** and DPP4T, the dopant would attract electrons in DPP4T more or less. In the meanwhile, the IP of 4.69 eV for DPP4T was the electron-donating ability of a single DPP4T chain with limited repeat units. The IP for DPP4T films at the well-packing region would be lower than 4.69 eV. Therefore, although the dopant might not be able to completely accept an electron from the DPP4T, it could dope DPP4T by affecting the electron distribution in the DPP4T films, thus resulting in low electrical conductivity. Secondly, **A1** was a well-known Lewis acid, which might form a water-Lewis acid complex to promote the p-type doping of DPP4T as suggested by Nguyen and Jang *et al.* (**Nat. Mater.**, 2019, 18, 1327; **Adv. Energy Mater.**, 2020, 10, 2002521).

The slight doping of DPP4T by **A1** would not harm the major conclusion that the coupled reaction between two weak dopants (dopant and additive) could promote the doping efficiency of DPP4T.

Related contents have been added in the revised supporting information.

REVIEWER COMMENTS

Reviewer #1 (Remarks to the Author):

I think the authors were able to respond appropriately to the reviewers' comments, and the readability of the manuscript has improved considerably.

Although I am still skeptical about the novelty of this research, I agree with publishing it, taking into consideration the opinions of other reviewers. However, there is still some ambiguity regarding the doping level calculation and the polarity of the Seebeck coefficient that I pointed out.

Hence, I would recommend that the sentences corresponding to this part should at least be excluded from the abstract.

Reviewer #3 (Remarks to the Author):

Referee #3 appreciates the authors responses to the various points raised. Some responses are satisfactory, two are not and require more discussion.

The first point is about the temperature stability of the doping. The authors provide a graph of conductivity vs. temperature, but no information is provided on the conditions of this experiment, in particular the time over which each temperature was held. Was it a few seconds, a minute, and hour? What were the conditions? Under N₂ in a glove box? In vacuum? Comparison with the temperature stability of other dopants reported in the literature would be very useful.

The second point concerns the doping of the polymer by A1. The explanations provided by the authors seem to show that the theoretical computations of EAs and IEs of the various materials presented here are rather inadequate. The referee is wondering whether the computations are limited to single molecules or polymer chains, neglecting polarization and other effects due to packing and molecular proximity. However, the EAs and IEs mentioned throughout the manuscript are presented as closely indicative of the ability of a certain dopant to dope or not to dope. And that is not correct. The authors should discuss this issue early on in the paper and clearly indicate that the theoretical IEs and EAs are limited in

accuracy and proposed as indicative only.

Reviewer #4 (Remarks to the Author):

Below, I provide my vision on how appropriate authors have addressed the concerns of Reviewer 2.

Reviewer 2 has raised eight questions, which have been thoroughly addressed by the authors, including performing additional measurements, adding clarification to the text. In particular,

The abstract has been revised to remove the potentially misleading statement that the method "overcomes the redox potential limitations in Marcus theory." Marcus theory considers the reaction between two species, whereas the doping process presented involves three reactants.

The authors have experimentally demonstrated that the results are not specific to the boron tetrafluoride anion. They show that doping is also effective when D1 (or D2, D3, D4) is replaced with D5 (TEMPO+PF6⁻). Specifically, the DPP4T-D5A1 system exhibits enhanced conductivity compared to DPP4T-D5. This evidence supports the universality of the results beyond the boron tetrafluoride anion.

The concern about the lack of detailed discussion on dopant-to-additive ratios and the apparent arbitrary selection of these ratios was addressed by adding more data points in Figures 3a and 3b and conducting additional experiments at low dopant molar ratios.

The authors have performed additional experiments to evaluate the EA and IP of the dopants and polymers using electrochemical methods. IP of DPP4T was also measured using UPS. These results agree well with previously reported literature values.

The authors provide additional explanations on why they think that species D and A find each other in the thin films, based on how they were prepared and past theoretical works. Ionic conductivity and ionic effects: the authors made AC impedance spectroscopy measurement to confirm that electronic transport was dominated over ionic effects. Seebach coefficient sign changes: the deep investigation of this effect is apparently out of scope of this work. Separate work is needed to explain this phenomenon.

The authors performed additional experiments to measure stability of the films, and found

out that the conductivity can be maintained > 85% for 7 days. However, the films have to be laminated by (PET) films to reach this stability.

The question about the absence of the signal in the EPR spectroscopy data was addressed by performing additional experiments. It was hypothesized that the TEMPO had been evaporated in the dry films.

Author responses suggest that all concerns of Reviewer 2 were appropriately addressed.

Below I provide my own comments, which the author may want to consider for their future work (the present work, to my opinion, is scientifically sound and comply scientific standards in its present view) and also report several mechanistic errors:

Comment 1.

Concerning issue (4) of Reviewer 2:

The level of the quantum chemistry simulations specifically using IP/EA computed using B3LYP functional to justify the proposed mechanism of doping is not enough. According to the manuscript, the Gibbs free energy of the reaction with D, A and the polymer is -21.31 kJ mol⁻¹, which is about -95 meV. To justify the proposed doping mechanism, the accuracy of the methods used to compute the free Gibbs energy must be this smaller than 100 meV.

B3LYP accuracy for IP/EA is similar to other hybrid functionals. For organic acceptor molecules, reported accuracy metric of PBE0(delta-SCF), namely MAE [J. Chem. Theory Comput. 2016, 12, 2, 605–614], is greater than 200 meV/300 meV/500 meV for IP/EA/(IP-EA), respectively. B3LYP accuracy is expected to be of the same order. Therefore, even before performing B3LYP simulations, one can already say, these can neither support nor refute the conclusions made in this work. The fact that IP of DPP4T and EA B(C6F5)₃ differ by not more than 100 meV is a lucky coincidence. TEMPO(A1) EA is already 500 meV away from the experimental results.

Still, experimental measurements of IP/EA support the proposed mechanism.

How it may have been done? For oligomer molecules the GW method might be used, which

is more accurate than B3LYP. For A and D, ccSD(T) method should be computationally feasible, which is more accurate than the GW method. Both mentioned methods have to be used in the complete basis set limit.

Comment 2.

Comment 2.1

The statement: "For p-type doping, it requires that the electron affinity (EA) of the dopant matches or exceeds the ionization potential (IP) of the polymeric semiconductor" is not fully correct. It is known that in organic charge transfer salts or in molecularly doped organic semiconductors, the Coulomb interaction energy is known to "stabilize" (that is to reduce the energy of) the products of the redox reaction between the host (polymer in this case) and the dopant. Unlike the initial neutral molecules, the products of redox (doping) reactions are oppositely charged and therefore their interaction energy is negative. Normally, the values between -0.5 eV to -1.0 eV are assumed taking into account the dielectric permittivity of the polymer and the distance. So, EA(dopant) must be equal or larger than the SUM of IP(host) and $-|VC|$, where VC is the mentioned interaction energy. Experimental error of the measured EA and IP is of the same order as VC, therefore it is often omitted. In the paper [6] cited next to the mentioned sentence, the role of VC is actually correctly described. Also, few sentences later, authors also acknowledged the role of Coulomb interactions between doped polymers and counter-ions as an alternative way compared to further increasing of EA: "Attempts have been placed on exploring new strategies to mitigate this problem by tuning Coulomb interactions between the doped polymeric semiconductors and the counter ions", which is inconsistent with the initial statement.

Comment 2.2

Coulomb interaction between the products of the reaction (1).

From the description in the paper, one can assume that the Gibbs free energy is simply the difference of the adiabatic IP of the polymer and EA of the dopant. However, it is very probable (especially at high molar dopant ratios) that the reactants and consequently

ionized reaction products namely positively charged polymer and dopants anions are in close proximity, which may make their coulomb interaction energy (several hundreds of meV) not negligible, especially compared to the resulted gibbs free energy of Eq. (3), around -100 meV.

Therefore, the right way to compute the free Gibbs energy of the reaction (1) is to subtract the assumed coulomb interaction of its products:

$$\Delta G_1 = IP - EA - |VC|$$

This will make the proposed mechanism even more thermodynamically favorable.

Comment 3.

Lines 326-329. Duplicated sentence:

“The doping efficiency induced electrical conductivity improvement has also been reported in previous works The doping efficiency induced electrical conductivity improvement has also been reported in previous works.”

Comment 4.

Equation 4 follows the sentence “The detailed screening of dopant-to-additive ratio was discussed in the supporting information.”, which puts Eq.4 out of the context.

Manuscript ID: NCOMMS-24-09903A

Title: Efficient molecular doping of polymeric semiconductors improved by coupled reaction

Reviewer Comments to Author:

Reviewer 1

General comments:

I think the authors were able to respond appropriately to the reviewers' comments, and the readability of the manuscript has improved considerably. Although I am still skeptical about the novelty of this research, I agree with publishing it, taking into consideration the opinions of other reviewers. However, there is still some ambiguity regarding the doping level calculation and the polarity of the Seebeck coefficient that I pointed out. Hence, I would recommend that the sentences corresponding to this part should at least be excluded from the abstract.

Authors' response:

Thank you very much for your kind help to improve the quality of this manuscript. We sincerely appreciate your time and efforts. The sentences corresponding to the doping level and polarity of the Seebeck coefficient have been removed from the abstract according to your suggestion. The revised abstract has been shown below:

“Chemical doping is an important way to tune the electrical properties of the polymeric semiconductors for modern organic electronics. However, the doping mechanism is far from being well-understood. Exploring new chemical doping method to improve the electrical conductivity of polymers is still very attractive for researchers. In this work, we report a newly developed method of doping a polymer semiconductor aided by the coupled reaction that commonly exists in biological systems where a non-spontaneous reaction is driven by a spontaneous reaction. During the doping process, the chemical reaction between the dopant and the polymer is promoted by introducing a thermodynamically favorable reaction via adding additives that are highly reactive to the reduction product of the dopant to form a coupled reaction, thus significantly improving the electrical conductivity of polymers by 3-7 orders. Theoretical calculation indicates that neither the dopant nor the additive has sufficiently large electron affinity to dope the polymer with a high ionization energy. However, efficient p-doping has been achieved because the addition of the dopant and the additive together leads to a complex that can accept one (or eventually two) electron from the polymer. This coupled reaction doping process shows the potential of wide applications in exploring new doping systems to prepare functional conducting polymers, which could be a powerful tool for modern organic electronics.”

Related contents have been added in the revised manuscript.

Reviewer 3

General comments:

Referee #3 appreciates the authors responses to the various points raised. Some responses are satisfactory, two are not and require more discussion.

Authors' response:

Thank you very much for helping us to improve the quality of this manuscript. We revised the paper according to your previous valuable comments. We appreciate any further suggestions and comments.

Reviewer's Comments 1:

The first point is about the temperature stability of the doping. The authors provide a graph of conductivity vs. temperature, but no information is provided on the conditions of this experiment, in particular the time over which each temperature was held. Was it a few seconds, a minute, and hour? What were the conditions? Under N₂ in a glove box? In vacuum? Comparison with the temperature stability of other dopants reported in the literature would be very useful.

Authors' response:

Thank you very much for your good suggestion. We are sorry for the missing description of the experimental conditions for the temperature stability of the doping, which have been added in the revised supporting information as shown below:

“The temperature dependent electrical conductivity of DPP4T-**D2A1** films at dopant molar ratio of 0.3 in **Figure S8a** was obtained with the commercial equipment NETZSCH SBA-458 (Germany). The electrical conductivities were measured with a four-probe method under argon protection. The temperature rise rate was 0.5 K/min. The total measurement time was about 3 h while increasing the temperature from 25°C to 120°C.

The electrical conductivity varying with the doping temperature in **Figure S8b** was measured with DPP4T-**D2A1** films that were obtained by drop-casting a mixture of polymer, dopant and additive (dopant molar ratio is 0.3) after stirring and reacting at different temperatures for 2 hours under N₂ protection in the glove box. After the films dried at room temperature under N₂ protection in the glove box, their electrical conductivities were measured with NETZSCH SBA-458 (Germany) by using four-probe method under argon protection at room temperature.”

Comparison with the temperature stability of other dopants reported in the literature has been provide as follows:

“The obtained p-doping films in this work exhibited a good thermal stability, which was better than the high EA value dopant FeCl₃ doped polymer (**Adv. Mater.** 2022, 34,

2102988; **Adv. Mater.** 2022, 34, 2106624; **CCS Chem.** 2021, 3, 2482). The electrical conductivity of the FeCl₃ doped polymer decreased to ~50% of the electrical conductivity measured at room temperature after being kept at 60°C (**Adv. Mater.** 2022, 34, 2102988). The thermal stability was comparable to the low EA value dopant F4TCNQ (2,3,5,6-tetrafluoro-7,7,8,8-tetracyanoquinodimethane) doped polymers (**Adv. Mater.** 2017, 29, 1700930). The results indicated that the coupled reaction doping method was promising for the preparation of efficiently doped and temperature stable conducting polymers for organic electronics by avoiding the utilization of high EA value dopants.”

Related contents have been added in the revised supporting information.

Reviewer's Comments 2:

The second point concerns the doping of the polymer by A1. The explanations provided by the authors seem to show that the theoretical computations of EAs and IEs of the various materials presented here are rather inadequate. The referee is wondering whether the computations are limited to single molecules or polymer chains, neglecting polarization and other effects due to packing and molecular proximity. However, the EAs and IEs mentioned throughout the manuscript are presented as closely indicative of the ability of a certain dopant to dope or not to dope. And that is not correct. The authors should discuss this issue early on in the paper and clearly indicate that the theoretical IEs and EAs are limited in accuracy and proposed as indicative only.

Authors' response:

Thank you very much for your insightful suggestions. A note has been provided at the beginning of the discussion of the theoretical calculation, which is shown below:

“It should be noted that both the EA and the IP values were only used for qualitatively understanding the doping process since these values were calculated with a single molecule or an oligomer under ideal environmental conditions without considering other effects such as the molecular packing and the molecular proximity, *etc* (**Adv. Mater.**, 2017, 29, 1703063; **Angew. Chem. Int. Ed.** 2021, 60; **Adv. Mater.** 2023, 35, 2300634). Therefore, the theoretical EA and IP values were proposed as indicative only due to the limitation in accuracy from practical conditions.”

In the meanwhile, the introduction has been revised to avoid the misunderstanding of the role of EA and IP values. The revised text is shown below:

“Although the electron-transfer process is very complex that can be affected by many factors such as the dopant-polymer miscibility (**Nat. Commun.**, 2020, 11, 3292), the packing order of the polymer chains (**Nat. Mater.**, 2013, 12, 719), *etc*, in many cases, the electron affinity (EA) and the ionic potential (IP) values are still used as an important

indicative to determine whether the efficient electron transfer (doping) occurs between the dopant and the polymer (**Nature**, 2021, 599, 67). Many researchers believe that dopants with a larger EA than the IP of polymers will be favorable for efficient electron transfer from the polymer to the dopant in p-type doping (**J. Phys. Chem. C**, 2024, 128, 1258; **Chem. Rev.**, 2016, 116, 13714; **Adv. Electron. Mater.**, 2022, 8, 2100888). Watanabe *et. al.* claims that the EA of the dopant should match or exceed the IP of the polymer for p-doping (**Nature**, 2019 572, 634). Moulé *et. al.* suggests that the EA of the dopant must be higher than the IP of polymer by ~ 0.11 eV for the efficient electron transfer to occur (**J. Phys. Chem. C**, 2024, 128, 1258).”

Related contents have been added in the revised manuscript.

Reviewer 4

General comments 1:

Below, I provide my vision on how appropriate authors have addressed the concerns of Reviewer 2. Reviewer 2 has raised eight questions, which have been thoroughly addressed by the authors, including performing additional measurements, adding clarification to the text. In particular. The abstract has been revised to remove the potentially misleading statement that the method "overcomes the redox potential limitations in Marcus theory." Marcus' theory considers the reaction between two species, whereas the doping process presented involves three reactants. The authors have experimentally demonstrated that the results are not specific to the boron tetrafluoride anion. They show that doping is also effective when D1 (or D2, D3, D4) is replaced with D5 (TEMPO+PF6-). Specifically, the DPP4T-D5A1 system exhibits enhanced conductivity compared to DPP4T-D5. This evidence supports the universality of the results beyond the boron tetrafluoride anion. The concern about the lack of detailed discussion on dopant-to-additive ratios and the apparent arbitrary selection of these ratios was addressed by adding more data points in Figures 3a and 3b and conducting additional experiments at low dopant molar ratios. The authors have performed additional experiments to evaluate the EA and IP of the dopants and polymers using electrochemical methods. IP of DPP4T was also measured using UPS. These results agree well with previously reported literature values. The authors provide additional explanations on why they think that species D and A find each other in the thin films, based on how they were prepared and past theoretical works. Ionic conductivity and ionic effects: the authors made AC impedance spectroscopy measurements to confirm that electronic transport was dominated over ionic effects. Seebeck coefficient sign changes: the deep investigation of this effect is apparently out of the scope of this work. Separate work is needed to explain this phenomenon. The authors performed additional experiments to measure the stability of the films and found that the conductivity can be maintained at > 85% for 7 days. However, the films have to be laminated by (PET) films to reach this stability. The question about the absence of the signal in the EPR spectroscopy data was addressed by performing additional experiments. It was hypothesized that the TEMPO had been evaporated in the dry films. Author responses suggest that all concerns of Reviewer 2 were appropriately addressed.

Authors' response:

Thank you very much for your valuable and supportive comments. We sincerely appreciate your time and efforts. We truly appreciate Reviewer 2' help to improve the quality of this manuscript as well.

General comments 1:

Below I provide my comments, which the author may want to consider for their future work (the present work, to my opinion, is scientifically sound and comply scientific standards in its present view) and also report several mechanistic errors.

Authors' response:

Thank you very much for your insightful suggestions. We have tried to revise this manuscript according to your valuable suggestions in this work. Your suggestions will be seriously considered in our future work as well. The mechanistic errors have been corrected. We sincerely appreciate your time and efforts.

Reviewer's Comments 1:

The level of the quantum chemistry simulations specifically using IP/EA computed using B3LYP functional to justify the proposed mechanism of doping is not enough. According to the manuscript, the Gibbs free energy of the reaction with D, A, and the polymer is $-21.31 \text{ kJ mol}^{-1}$, which is about -95 meV . To justify the proposed doping mechanism, the accuracy of the methods used to compute the free Gibbs energy must be smaller than 100 meV . B3LYP accuracy for IP/EA is similar to other hybrid functionals. For organic acceptor molecules, the reported accuracy metric of PBE0(Δ -SCF), namely MAE [J. Chem. Theory Comput. 2016, 12, 2, 605–614], is greater than $200 \text{ meV}/300 \text{ meV}/500 \text{ meV}$ for IP/EA/(IP-EA), respectively. B3LYP accuracy is expected to be of the same order. Therefore, even before performing B3LYP simulations, one can already say, that these can neither support nor refute the conclusions made in this work. The fact that the IP of DPP4T and EA B(C6F5)₃ differ by not more than 100 meV is a lucky coincidence. TEMPO(A1) EA is already 500 meV away from the experimental results. Still, experimental measurements of IP/EA support the proposed mechanism. How it may have been done? For oligomer molecules, the GW method might be used, which is more accurate than B3LYP. For A and D, the ccsd(t) method should be computationally feasible, which is more accurate than the GW method. Both mentioned methods have to be used in the complete basis set limit.

Authors' response:

Thank you very much for your good suggestion.

B3LYP/6-311+g(d,p) has an acceptable accuracy, which has been widely used in literatures to calculate the Gibbs free energy of organic semiconductors (**Nature** 2022, 611, 271; **Nature** 2021, 599, 67). In this work, Grimme's D3 dispersion correction with BJ damping function was added to improve the accuracy of B3LYP according to literature (**Phys. Chem. Chem. Phys.** 2017, 19, 32184; **Chem. Rev.** 2016, 116, 5105). As suggested by Lu *et. al.*, B3LYP Grimme's D3 dispersion method was a cheap and relatively reliable method, which could provide qualitative calculation results consistent

with the accurate method CCDS(T) mentioned by the reviewer (**J. Mol. Model.** 2013, 19, 5387). The mean absolute deviation of B3LYP Grimme's D3 dispersion method in computed free energy was 10.5 kJ/mol, that was 108 meV, as reported by Ho *et. al.* (**J. Phys. Chem. A** 2021, 125, 9838). The accuracy should be good enough to tell the Gibbs free energy changes in this work which was smaller than -21.31 kJ/mol, that was -220 meV. Therefore, the theoretical calculation indicated that the coupled reaction was thermodynamically favorable.

We hope the reviewer could understand that the theoretical calculation values were only used for qualitatively understanding the doping process since these values were calculated with a single molecule or an oligomer under ideal environmental conditions without considering other effects such as the molecular packing and the molecular proximity, *etc* (**Adv. Mater.**, 2017, 29, 1703063; **Angew. Chem. Int. Ed.** 2021, 60; **Adv. Mater.** 2023, 35, 2300634). Therefore, it was hard to have all the theoretical values closely match the experimental values. A 500 meV difference between the theoretical and experimental values were commonly reported in previous literature (**Angew. Chem. Int. Ed.** 2024, 63, e202402642; **Angew. Chem. Int. Ed.** 2023, 62, e202219262; **Angew. Chem. Int. Ed.** 2024, 63, e202319658). In addition, the difference between the theoretical and experimental values would not change the main conclusion in this manuscript. We appreciate the reviewer's rigorous academic attitude and your suggestions will be seriously considered in our future work.

Related contents have been added in the revised manuscript.

Reviewer's Comments 2:

Comment 2.1

The statement: "For p-type doping, it requires that the electron affinity (EA) of the dopant matches or exceeds the ionization potential (IP) of the polymeric semiconductor" is not fully correct. It is known that in organic charge transfer salts or molecularly doped organic semiconductors, the coulomb interaction energy is known to "stabilize" (that is to reduce the energy of) the products of the redox reaction between the host (polymer in this case) and the dopant. Unlike the initial neutral molecules, the products of redox (doping) reactions are oppositely charged and therefore their interaction energy is negative. Normally, the values between -0.5 eV to -1.0 eV are assumed taking into account the dielectric permittivity of the polymer and the distance. So, EA(dopant) must be equal to or larger than the SUM of IP(host) and -|VC|, where VC is the mentioned interaction energy. Experimental error of the measured EA and IP is of the same order as VC, therefore it is often omitted. In the paper [6] cited next to the mentioned sentence, the role of VC is actually correctly described. Also, a few sentences later, the authors also acknowledged the role of Coulomb interactions between doped polymers

and counter-ions as an alternative way compared to further increasing of EA: “Attempts have been placed on exploring new strategies to mitigate this problem by tunings Coulomb interactions between the doped polymeric semiconductors and the counter ions”, which is inconsistent with the initial statement.

Authors’ response:

Thank you very much for your good suggestion.

We agree with the reviewer that the statement of “For p-type doping, it requires that the electron affinity (EA) of the dopant matches or exceeds the ionization potential (IP) of the polymeric semiconductor” is not fully correct. To avoid the misunderstanding of readers, it has been replaced in the revised manuscript with the sentences shown below:

“Molecular doping occurs through transferring electrons from the polymeric semiconductors to the added molecular dopants (for example, p-type doping). Although the electron-transfer process is very complex that can be affected by many factors such as the dopant-polymer miscibility (**Nat. Commun.**, 2020, 11, 3292), the packing order of the polymer chains (**Nat. Mater.**, 2013, 12, 719), *etc*, in many cases, the electron affinity (EA) and the ionic potential (IP) values are still used as an important indicative to determine whether the efficient electron transfer (doping) occurs between the dopant and the polymer (**Nature**, 2021, 599, 67). Many researchers believe that dopants with a larger EA than the IP of polymers will be favorable for efficient electron transfer from the polymer to the dopant in p-type doping (**J. Phys. Chem. C**, 2024, 128, 1258; **Chem. Rev.**, 2016, 116, 13714; **Adv. Electron. Mater.**, 2022, 8, 2100888). Watanabe *et. al.* claims that the EA of the dopant should match or exceed the IP of the polymer for p-doping (**Nature**, 2019 572, 634). Moulé *et. al.* suggests that the EA of the dopant must be higher than the IP of polymer by ~ 0.11 eV for the efficient electron transfer to occur (**J. Phys. Chem. C**, 2024, 128, 1258).”

Related contents have been added in the revised manuscript.

Reviewer's Comments 2:

Comment 2.2

Coulomb interaction between the products of the reaction (1). From the description in the paper, one can assume that the Gibbs free energy is simply the difference of the adiabatic IP of the polymer and EA of the dopant. However, it is very probable (especially at high molar dopant ratios) that the reactants and consequently ionized reaction products namely positively charged polymer and dopants anions are in close proximity, which may make their coulomb interaction energy (several hundreds of meV) not negligible, especially compared to the resulted Gibbs free energy of Eq. (3), around -100 meV. Therefore, the right way to compute the free Gibbs energy of the reaction (1) is to subtract the assumed column interaction of its products:

$$\Delta G_1 = IP - EA - |VC|$$

This will make the proposed mechanism even more thermodynamically favorable.

Authors' response:

Thank you very much for your good suggestion.

Considering all the calculated values were obtained with a single molecule or an oligomer under ideal environmental conditions without considering other effects such as the molecular packing and the molecular proximity, *etc* (**Adv. Mater.**, 2017, 29, 1703063; **Angew. Chem. Int. Ed.** 2021, 60; **Adv. Mater.** 2023, 35, 2300634), they were proposed as indicative only due to the limitation in accuracy from practical conditions. Furthermore, we agree with the reviewer that the consideration of the column interaction of the products would make the proposed mechanism even more thermodynamically favorable. The above accurate free Gibbs energy calculation will be considered in our future work.

Related contents have been added in the supporting information.

Reviewer's Comments 3:

Lines 326-329. Duplicated sentence: "The doping efficiency induced electrical conductivity improvement has also been reported in previous works The doping efficiency induced electrical conductivity improvement has also been reported in previous works."

Authors' response:

Thank you very much for your suggestion. The duplicated sentence has been removed from manuscript.

Related contents have been corrected in the revised manuscript.

Reviewer's Comments 4:

Equation 4 follows the sentence "The detailed screening of dopant-to-additive ratio was discussed in the supporting information.", which puts Eq.4 out of the context.

Authors' response:

Thank you very much for pointing this out. We are sorry for the mistake. The related context has been re-written as shown below:

"..... The detailed screening of dopant-to-additive ratio was discussed in the supporting information.

The equation (4) shows that TEMPO⁺ accepts two electrons one by one to produce TEMPO⁻.

where TEMPO[·] is the radical status of TEMPO and TEMPO⁻ is the TEMPO anion
(**Angew. Chem. Inter. Ed.**, 2011, 50, 5034; **Chem. Rev.**, 2023, 123, 10302).....”

Related contents have been corrected in the revised manuscript.

REVIEWERS' COMMENTS

Reviewer #3 (Remarks to the Author):

With this second round of revisions, the authors have addressed all the referee's concerns satisfactorily. The manuscript can be accepted for publication.

Reviewer #4 (Remarks to the Author):

My comments and suggestions have been addressed. From my perspective, the manuscript can now be accepted.

Reviewer Comments to Author:

Reviewer #3 (Remarks to the Author):

With this second round of revisions, the authors have addressed all the referee's concerns satisfactorily. The manuscript can be accepted for publication.

Authors' response:

Thanks a lot. We truly appreciate your help in improving the quality of this manuscript.

Reviewer #4 (Remarks to the Author)

General comments:

My comments and suggestions have been addressed. From my perspective, the manuscript can now be accepted.

Authors' response:

Thanks a lot. We truly appreciate your help in improving the quality of this manuscript.